# A long-term (2002 to 2017) record of closed-path and open-path eddy covariance $CO_2$ net ecosystem exchange fluxes from the Siberian Arctic

David Holl[1], Christian Wille[2], Torsten Sachs[2], Peter Schreiber[3], Benjamin R.K. Runkle[4], Lutz Beckebanze[1], Moritz Langer[3], Julia Boike[3,8], Eva-Maria Pfeiffer[1], Irina Fedorova[5], Dimitry Yu. Bolshianov[6], Mikhail N. Grigoriev[7], and Lars Kutzbach[1]

[1]Institute of Soil Science, Center for Earth System Research and Sustainability (CEN), Universität Hamburg, Hamburg, Germany
[2]Helmholtz-Zentrum Potsdam – Deutsches GeoForschungsZentrum (GFZ), Potsdam, Germany
[3]Alfred Wegener Institute Helmholtz Centre for Polar and Marine Research, Potsdam, Germany
[4]Department of Biological & Agricultural Engineering, University of Arkansas, Fayetteville, USA
[5]Saint Petersburg State University – Institute of Earth Sciences, St. Petersturg, Russia
[6]Arctic and Antarctic Research Institute, St. Petersburg, Russia
[7]Permafrost Institute, Yakutsk, Russia
[8]Humboldt-Universität zu Berlin, Geography Department, Berlin, Germany

**Correspondence:** David Holl (david.holl@uni-hamburg.de)

**Abstract.** Ground-based observations of land–atmosphere fluxes are necessary to progressively improve global climate models. Observed data can be used for model evaluation and to develop or tune process models. In arctic permafrost regions, climate–carbon feedbacks are amplified. Therefore, increased efforts to better represent these regions in global climate models have been made in recent years. We present a multiannual time series of land–atmosphere carbon dioxide fluxes measured *in situ* with the eddy covariance technique in the Siberian Arctic (72° 22' N, 126° 30' E). The site is part of the international network of eddy covariance flux observation stations (FLUXNET, Site ID: Ru-Sam). The data set includes consistently processed fluxes based on concentration measurements of closed-path and open-path gas analyzers. With parallel records from both sensor types, we were able to apply a site-specific correction to open-path fluxes. This correction is necessary due to a deterioration of data, caused by heat generated by the electronics of open-path gas analyzers. Parameterizing this correction for subperiods of distinct sensor setups yielded good agreement between open and closed-path fluxes. We compiled a long-term (2002 to 2017) carbon dioxide flux time series that we additionally gap-filled with a standardized approach. The data set was uploaded to the Pangaea data base and can be accessed through https://doi.pangaea.de/10.1594/PANGAEA.892751.

# 1 Introduction

The release of the Arctic's belowground carbon (C) pools to the atmosphere can potentially act as a positive feedback on climate change. Organic material that is now stored in the permanently frozen soil and largely inaccessible for microbial decomposition might become available under a warming climate resulting in an increased release of greenhouse gases from Arctic regions (Schuur et al., 2015). At the same time, the Arctic vegetation responds to ongoing warming with a greening trend (Park et al., 2016), probably enhancing summer carbon assimilation. Although the importance of permafrost carbon pools for a potential amplification of climate change has been widely recognized (e.g. Zimov et al., 2006; Davidson and Janssens, 2006; Schuur et al., 2008; Khvorostyanov et al., 2008; Tarnocai et al., 2009; Koven et al., 2011; Schneider von Deimling et al., 2012; MacDougall et al., 2012; Schuur et al., 2013; McGuire et al., 2018), the earth system models analyzed for the Fifth Assessment Report (AR5) of the Intergovernmental Panel on Climate Change (IPCC) did not include permafrost carbon emissions.

While efforts to include permafrost dynamics into global climate models have been made recently (e. g. Wania et al., 2009a, b, 2010; Ekici et al., 2014; Kaiser et al., 2017; McGuire et al., 2018), models can be improved by using ground-based flux measurements for calibration and validation. McGuire et al. (2012) assessed the carbon balance of the Arctic tundra combining ground-based observations, process and atmospheric inversion models. The authors found that the uncertainty with which a carbon balance can be quantified is still very large, with upper and lower uncertainty bounds indicating the Arctic tundra as a sink for carbon at one and as a C-source at the other bound. McGuire et al. (2012) conclude that reducing uncertainties of regional estimates based on observational data relies on high quality ground-based measurements that should be placed strategically, e. g. along hydrological or vegetation gradients. *In situ* gas flux measurements from the Arctic are, however, still scarce. Moreover, the available data is biased towards Alaska, observations from the Eurasian Arctic are even more scarce (Oechel et al., 2014). To be able to distinguish climate change-related flux responses from interannual variability, long-term data sets are essential as recently argued by Baldocchi et al. (2017).

Within the scope of this publication, we aimed at creating a high quality, long-term $CO_2$ flux data set from a polygonal tundra site in the Russian Arctic. We had the opportunity to analyze a 16 year record of eddy covariance data that includes periods with simultaneous measurements from two different (closed-path and open-path) $CO_2$ gas analyzer types. Our objective was to consistently process the data while following standardized quality control methods to allow for comparability between the different years of our record and with other data sets. We additionally aimed at cross-calibrating open-path and closed-path $CO_2$ fluxes and at gap-filling the data set by employing the method of Reichstein et al. (2005) that is widely used in the FLUXNET community.

# 2 Site description

The investigation site is located on Samoylov Island in the southern central part of the Lena River Delta at 72° 22' N, 126° 30' E (see Figure 1). The fan-shaped delta covers an area of roughly 30000 $km^2$ (Grigoriev, 1993; Schneider et al., 2009) and is characterized by a network of channels and more than 1500 islands (Antonov, 1967). Being the largest delta in the Arctic and one of the largest worldwide (Walker, 1998), it lies in the continuous permafrost zone with permafrost depths of about 500

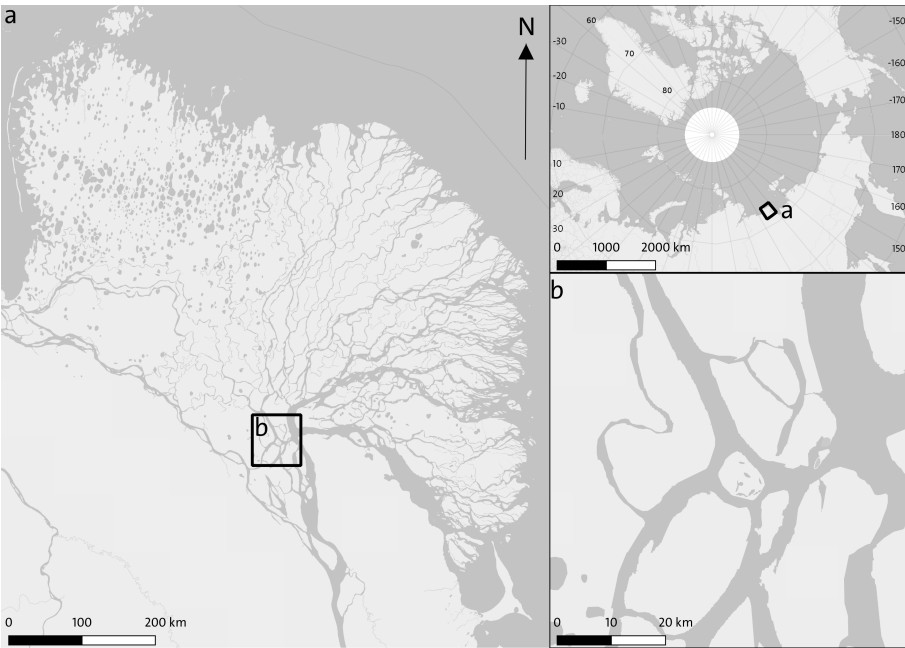

**Figure 1.** Location of Samoylov Island (center of panel b) in the Lena River Delta (panel a). Map data from: OpenStreetMap contributors, under Open Database License

to 600 m (Romanovskii et al., 2004; Yershov, 2004; Brown et al., 1997). Mean annual permafrost temperatures range around -9 °C at 10 m depth (Romanovsky et al., 2010), making the Lena River Delta one of the coldest permafrost regions on earth. Boike et al. (2013) inferred an annual mean soil temperature of -8.6°C at 10.7 m depth from a 2006 to 2011 time series of temperature measurements in a borehole on Samoylov Island. Based on long-term hydrological observations in the delta area,
Fedorova et al. (2015) found an increase in discharge as well as in sediment flux indicating recently intensified thawing of ice complex sediments in the region.

Grigoriev (1993) divides the delta area in three main geomorphological units. The oldest, ice-rich river terrace consists of fine-grained sediments with high organic content. It developed as an eroded Pleistocene plane characterized by polygonal ground and thermokarst processes. The second largest unit consists of Late Pleistocene to Early Holocene sandy sediments with
10 low ice content and covers 23 % of the north-western part (Schneider et al., 2009). Samoylov Island is part of the third unit, the Mid to Late Holocene river terrace (Bolshiyanov et al., 2015), which makes up about two thirds of the delta (Schwamborn et al., 2002).

The island itself consists of two morphological units, an annually flooded, modern floodplain (1.49 $km^2$) in the west and a Late Holecene river terrace (2.85 $km^2$) in the east, which lies 10 to 16 m a.s.l. and is not flooded regularly (Kutzbach et al., 2007;
Boike et al., 2013). The data presented here was collected with eddy covariance systems installed on the elevated river terrace. In contrast to the modern floodplain, the river terrace's surface is patterned due to frost-action that formed a wet polygonal tundra landscape consisting of mostly low-centered and some high-centered ice-wedge polygons as well as thermokarst lakes

and channels. Due to the underlying permafrost and thereby hampered drainage, water-saturated soils or ponds form in the polygon centers, whereas on the rims, which can be elevated up to 50 cm above the centers, a drier, moderately moist water regime prevails (Kutzbach et al., 2007; Helbig et al., 2013). Accordingly, the vegetation community in the wetter centers is dominated by hydrophytic sedges (*Carex aquatilis*, *Carex chordorrhiza*, *Carex rariflora*) and mosses (e. g. *Limprichtia*

*revolvens, Meesia longiseta, Aulacomnium turgidum*). Mesophytic dwarf shrubs (e. g. *Dryas octopetala, Salix glauca*), forbs (e. g. *Astragalus frigidus*) and mosses (e.g. *Hylocomium splendens, Timmia austriaca*) dominate on the rims (Kutzbach et al., 2004; Pfeiffer and Grigoriev, 2002). Maximum summer leaf coverage was estimated by Kutzbach et al. (2004) to be 0.3 for vascular plants and 0.95 for mosses and lichens at both polygon centers and rims. The river terrace as a whole is composed of polygon rims with a coverage of 60 to 65 % and of depressed surfaces (including vegetated and water filled polygon centers

as well as lakes and channels) that cover the remaining 35 to 40 % of area (Kutzbach et al., 2007; Sachs et al., 2010; Muster et al., 2012; Boike et al., 2013).

An arctic-continental climate with low mean annual temperatures prevails in the Lena River Delta. Although precipitation is low as well, the climate can be considered humid as evaporation rates are low due to low ambient temperatures and relative humidity is high (Kutzbach, 2006; Boike et al., 2008; Langer et al., 2011a, b). Based on long-term (1998 to 2017) *in situ*

measurements on Samoylov Island, Boike et al. (2018) inferred an annual mean air temperature of -12.3 °C; the coldest and warmest months being February and July with mean temperatures of -32.7 °C and 9.5 °C respectively. For the period from 1998 to 2011, Boike et al. (2013) estimated total annual precipitation to be composed of $124 \pm 57$ mm summer rainfall and $65 \pm 35$ mm snowfall. Interannual variability in rainfall was, however, very high, with a maximum of 199 mm and a minimum of 48 mm. Snow melt usually starts in mid-May and lasts until early June. Snow accumulation typically commences between late

September and early October. Between 1998 and 2011, the snow season lasted on average $224 \pm 18$ days, the snow-free period $138 \pm 18$ days. Snow depth was reported by Boike et al. (2018) averaging 0.3 m between 2002 and 2017 with a maximum of 0.8 m in 2017. Beginning in early to mid-June, the soil starts to thaw from the top, forming the so called active layer. Boike et al. (2013) report a mean active layer depth in August of 49 cm with a maximum of 79 cm between 1998 and 2011. The closest WMO (World Meteorological Organisation) weather station is located on the continent, around 110 km southeast from

Samoylov Island in the city of Tiksi (WMO ID 21824). Between 1936 and 2017 the mean air temperature reported from Tiksi is - 12.74 °C, mean annual precipitation amounts to 304.5 mm (AARI, 2018). While the mean air temperature in Tiksi is very similar to the 20-year mean from Samoylov Island, average annual precipitation appears to be much higher in Tiksi than in the delta region. Boike et al. (2013) explain this divergence with the fact that Tiksi is located at the coast of the Laptev sea and surrounded by mountains.

The soils of Samoylov Island were classified as *Gelisols* by Zubrzycki et al. (2013) based on work by Pfeiffer and Grigoriev (2002) according to the US Soil Taxonomy (Soil Survey Staff, 2014). On subgroup level, typical soils of the river terrace are *Glacic Aquiturbels*, which developed on the polygon rims and are characterized by the translocation of soil material due to freeze-thaw processes (cryoturbation). In the wetter polygon centers *Typic Historthels* formed. On the more sand-rich active floodplain, *Typic Aquorthels* and *Typic Psammorthels* dominate. According to the FAO World Reference Base for Soil

Ressources (IUSS Working Group WRB, 2015), the diverse soils of Samoylov Island belong to the reference soil group of

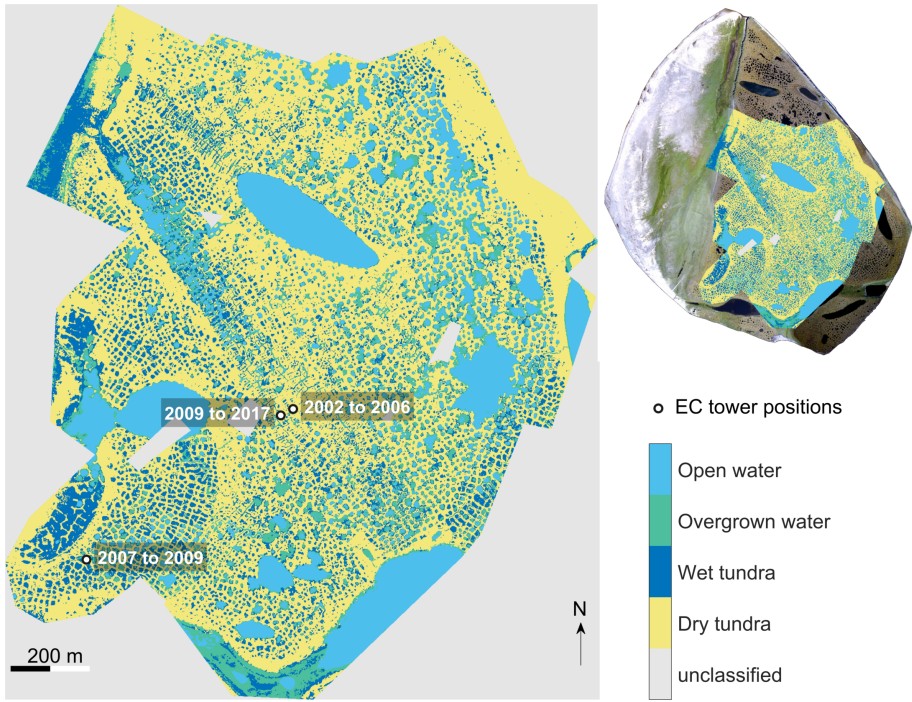

**Figure 2.** Eddy covariance (EC) tower positions on the river terrace of Samoylov Island and surface class distribution according to Muster et al. (2012). Photographic image of the entire island (top right corner) from Boike et al. (2012).

*Cryosols*. Zubrzycki et al. (2013) estimated the soil organic carbon (SOC) stocks for the upper meter of the island's two major landscape units to be $29 \pm 10 \, \mathrm{kg \, m^{-2}}$ for the river terrace and $14 \pm 7 \, \mathrm{kg \, m^{-2}}$ for the active floodplain.

## 3 Methods

### 3.1 Instrumentation

5 We used the eddy covariance (EC) technique to determine half-hourly gas and energy fluxes. The EC method requires high frequency (typically $> 10 \, \mathrm{Hz}$) raw gas concentration and three-dimensional wind velocity measurements. A comprehensive description of the EC approach is given for example by Aubinet et al. (2012). We recorded carbon dioxide ($CO_2$) and water vapor concentrations as well as three-dimensional wind velocity with changing instrumentation on three different tower structures, all located on the central river terrace of Samoylov Island between 2002 and 2017 (see Figure 2). We deployed open-path (OP) as

10 well as closed-path (CP) gas analyzers, at times simultaneously. Models, manufacturers and years of deployment are given in Table 1. Between the different setups, CP intake tube lengths varied from 5 to 8 m. OP analyzers were always installed inclined by about 10 degrees from the vertical, as suggested in the analyzer manuals. Raw data was recorded at 20 Hz except for the periods 22 August 2009 to 19 July 2010 (10 Hz) and 31 August 2012 to 17 May 2013 (5 Hz). Until 29 April 2014, all raw

data were recorded on a CR3000 data logger (Campbell Scientific, UK). From then on, CP analyzer and anemometer data were logged on a CR3000 whereas OP analyzer and anemometer data were recorded on a LI-7550 data logger (Licor Biosciences, USA). Although data coverage is biased towards the growing season, the data set contains considerably more shoulder season and winter fluxes in its second half from 2010 to 2017(see Table 1). The also increasing availability of year-round ancillary meteorological data resulted in gap-filled flux time series covering each half hour of the two years 2010 and 2016 (see Figure 4).

## 3.2  Flux processing

### 3.2.1  Prior considerations

Due to the contrasting designs of OP and CP analyzers, these sensor types have distinct signal response characteristics that we considered during data processing. The most apparent constructional difference between OP and CP gas analyzers is the presence or absence of a housing for the measurement cell that contains the optical path. In a CP instrument, the measurement cell is housed whereas the optical path of an OP analyzer is exposed to the atmosphere. CP systems are typically more bulky and installed at the base of an EC tower, from where tubing leads to an intake close to the anemometer. Sample air is drawn into the cell with a pump. OP sensors are commonly installed in close proximity to the anemometer and do not require a pump that greatly reduces the power consumption of OP instruments compared to CP setups. Due to the tubing acting as a low-pass filter, the response to high-frequency concentration variations is systematically attenuated in CP setups, as opposed to OP systems (Ibrom et al., 2007a). Moreover, the severity of frequency dampening can vary non-linearly with environmental conditions, especially with relative humidity (Runkle et al., 2012).

Infrared gas analyzers typically measure gas densities and report the number of molecules per volume of air. To be able to refer the mass of a gas to the mass of air, gas densities are transformed to mixing ratios using air density. However, as the optical path of an OP gas analyzer is exposed to the varying temperature, pressure and humidity conditions of the atmosphere, air density in the measurement cell fluctuates mainly due to thermal expansion/contraction and water dilution/concentration. This effect, that leads to faulty concentration readings of OP instruments and thereby to incorrect flux estimates, has first been described by Webb et al. (1980). The authors proposed two flux correction terms to compensate for these density fluctuation effects that are referred to as Webb-Pearman-Leuning (WPL) terms and have since been verified experimentally and theoretically and are routinely applied in OP EC studies. Especially at times of low gas fluxes, WPL terms can become orders of magnitude larger than raw gas fluxes (Munger et al., 2012). CP analyzers have the advantage of controlled temperature and pressure conditions in the measurement cell, allowing for the sample-wise calculation of mixing ratios rather than molar densities (Ibrom et al., 2007b) and thereby avoiding the need to apply air density fluctuation correction terms after raw flux calculation.

Major drawbacks of OP instruments, especially in harsh environments, are (1) their downtime during adverse weather conditions (e. g. precipitation) and (2) flux biases due to sensor self-heating (Burba et al., 2006, 2008). The OP self-heating effect was first recognized (Burba et al., 2006) due to apparent off-season $CO_2$ uptake in flux time series obtained with LI-7500 (LI-COR Biosciences, USA) OP gas analyzers. However, Kittler et al. (2017) recently found that this effect is not limited to

cold conditions but extends throughout all seasons. The necessary corrections can be substantial but decrease largely when the sensor is not mounted vertically but inclined instead as shown by Rogiers et al. (2008) and Järvi et al. (2009).

### 3.2.2 Processing steps

We performed separate flux processing steps on OP and CP data sets and computed half-hourly fluxes using the software EddyPro (Licor Biosciences, USA). An overview of the processing steps is given in Table 2. We detected and removed raw data spikes according to Vickers and Mahrt (1997), with a maximum of 1 % accepted spikes and a maximum of three samples as consecutive outliers. We applied an angle of attack correction, i.e. compensation for flow distortion induced by the anemometer frame (Nakai et al., 2006), on wind velocity data collected with the R3 (Gill Instruments Ltd., UK) anemometer. The majority of the wind velocity records come, however, from a CSAT3 (Campbell Scientific, UK) instrument for which this correction is not necessary. Coordinate rotation to align the anemometer x-axis to the current mean streamlines was calculated as double rotation according to Kaimal and Finnigan (1994). For OP fluxes, we compensated for air density fluctuations due to thermal expansion/contraction and water dilution/concentration following Webb et al. (1980). Because simultaneous water vapor concentration, cell temperature and cell pressure measurements from inside the CP analyzer were available, $CO_2$ concentrations from this sensor could be converted directly into mixing ratios, i.e. concentrations referring to dry air of constant temperature (Ibrom et al., 2007b; Burba et al., 2012), making further corrections for density fluctuations unnecessary. We compensated CP time lags by using the automatic timelag optimization option in EddyPro. For this procedure, prior to processing the complete data set, time lags were determined for a subperiod of raw data by covariance maximization (Fan et al., 1990). A searching window around the median of the found time lags (nominal timelag, $T_{nom}$) is defined by $T_{nom} \pm 3.5 \times MAD$, where $MAD$ is the median absolute deviation of the found time lags. When processing the complete data set, EddyPro performed a covariance maximization of vertical wind velocity and the scalar of interest for each half hour and then checked, whether the found time lag fell within the searching window defined before. If not, $T_{nom}$ was used as time lag. Water vapor concentration time series were binned in ten RH-classes, and the procedure was applied to each class, resulting in ten different nominal time lags. $CO_2$ concentrations were not binned in humidity-classes. We computed CP time lag statistics annually and within a year if pump speeds or instrumental setups varied. OP time lags were determined by covariance maximization within a searching window of -10 to 10 seconds. We evaluated OP time lags statistics, binned in classes of wind direction sectors, later on in the course of quality filtering.

Spectral attenuation in the high- and the low-frequency spectral range was compensated according to the following methods. Low-frequency signal loss due to the finite averaging time used for flux calculations (30 minutes) and due to linear raw data detrending was corrected for following the method of Moncrieff et al. (2004) for both OP and CP fluxes. High-frequency signal loss of OP fluxes due to path and volume averaging of the sonic anemometer and the gas analyzers as well as due to the separation between the two instruments were corrected for with the analytical approach of Moncrieff et al. (1997). High-frequency signal loss of CP fluxes due to spectral attenuation by the intake tube and volume averaging in the measurement cell were corrected for using the *in situ* method of Ibrom et al. (2007a). For each measurement period with a unique instrumental setup and CP pump speed, we determined the cut-off frequency of a first-order low-pass filter from ensemble means of 30-minute power

spectra of $CO_2$ concentration and sonic temperature time series data. The spectral correction factor was then parametrized as a function of the cut-off frequency found and the mean wind speed for stable and unstable atmospheric conditions as described by Ibrom et al. (2007a). Before using them for ensemble spectra estimations, the 30-minute power spectra were quality-filtered by applying the scheme of Vickers and Mahrt (1997), and by omitting half-hours that were assigned quality class 2 accord-

ing to Mauder and Foken (2004). High frequency noise was removed from the ensemble means of $CO_2$ concentration power spectra before the determination of the cut-off frequency where it was deemed necessary. High-frequency signal losses due to crosswind and vertical separation of the sample air tube intake and the anemometer were corrected for according to Horst and Lenschow (2009).

## 3.3   Quality filtering

We set EddyPro to calculate quality flags according to Mauder and Foken (2004) that represent flux quality in three classes (0, 1 and 2) with 0 denoting the highest and 2 denoting the lowest quality class. This quality evaluation is based on tests for stationarity and developed turbulence and thereby indicates whether general EC assumptions about atmospheric conditions were met during a flux calculation period. Flux quality assessment was largely based on the scheme of Mauder and Foken (2004). In the data set available for download, we included one column for each analyzer type containing this quality flag.

Additionally, we applied six further screening steps and flagged fluxes of low quality. If a flagged flux was not already assigned to class 2 according to Mauder and Foken (2004), we set the quality flag to 2. In our opinion, fluxes of quality class 2 should be omitted from further analysis. They are included in the reported data set for the sake of completeness. We performed the six additional flagging steps in the following sequence. An overview of these filtering steps including the number of flagged values is given in Tabele 3.

In **step 1**, skewness and kurtosis were computed with EddyPro for the half-hourly high frequency raw data time series of $CO_2$ concentration, vertical wind speed and sonic temperature. If any of these statistics was outside certain intervals (skewness: $[-2,2]$, kurtosis: $[1,8]$, equivalent to the hard flag defined by Vickers and Mahrt (1997)), $CO_2$ flux values were flagged.

In **step 2**, OP fluxes were additionally filtered for an instrument signal strength indication ($AGC$) recorded from the LI-7500 sensor. Along with a software upgrade, this diagnostic value was renamed to $RSSI$, and its definition was changed. We there-

fore recalculated the $AGC$ values for sensors not running on firmware version 6.6 and above (before July 2013). According to the old $AGC$ definition in the LI-7500 manual, typical clean window values range between 55 to 65 %. As dirt accumulates on the windows (or anywhere in the optical path), the $AGC$ value will increase up to 100 %. The new $RSSI$ value takes 100 % for clean windows and decreases as windows get dirtier. In order to obtain one consistent diagnostic variable for the cleanness of the optical path, $AGC$ was converted to the $RSSI$ range. $AGC$ values smaller than 44 were set to 44, then AGC values

were mapped to the RSSI range as follows.

$$RSSI(AGC) = 188 - 2 \cdot AGC \tag{1}$$

We flagged OP $CO_2$ flux values when $RSSI \leq 60$.

As quality control of the half-hourly time lag detection results was not applied during OP flux processing in EddyPro, we additionally screened OP time lags to identify low quality flux values in **step 3**. We divided the time lag data set into subsets of different instrumental setup, and binned the time lags of these subsets in 36 ten degree wind direction sectors. We used the 25th and 75th percentiles per class as filter thresholds. We flagged OP flux values with associated time lags outside the range spanned by these thresholds. Because we computed CP fluxes in EddyPro considering and compensating for low time lag detection quality, we did not perform this type of filtering step on CP fluxes.

In **step 4**, we flagged CP as well as OP fluxes when 30 minute average concentration measurements were larger than 450 ppm or smaller than 300 ppm. $CO_2$ concentrations outside this range indicate dirty OP gas analyzer optics or technical problems of the CP air sampling system (sudden pump speed changes due to brownouts, blocked filters, etc.).

To filter dubious, large OP fluxes that coincided with reasonable CP fluxes, we seleced all OP fluxes when simultaneously measured CP values ranged between -2 $\mu mol\,m^{-2}\,s^{-1}$ and 2 $\mu mol\,m^{-2}\,s^{-1}$. **Step 5** only affected OP data from this subset. We calculated the 99th and 1st percentile of this group and flagged fluxes from it when they lay outside this percentile range.

In **step 6**, we flagged remaining outliers in both the CP and OP data sets by using the 0.1st and 99.9th percentile (-3.5423 $\mu mol\,m^{-2}\,s^{-1}$ and 3.3473 $\mu mol\,m^{-2}\,s^{-1}$) of the CP time series after the concentration limits filter as absolute limits, to define an acceptable range of OP and CP flux values.

### 3.4   Open-path self-heating correction

To account for self-heating errors induced by the LI-7500 sensor electronics, we corrected OP fluxes as described by Kittler et al. (2017). The authors use WPL-corrected fluxes and add a correction term (Burba et al., 2006) that accounts for self-heating effects of vertically installed instruments. In their approach, Kittler et al. (2017) use a scaling factor $\xi$, taking values between 0 and 1, to trim the correction for inclined analyzer setups. With simultaneously available CP fluxes, we were able to estimate this scaling factor specifically for our site and periods of unique instrumental setups. As suggested by Kittler et al. (2017), we optimized this parameter with a nonlinear least squares method in Matlab (v. 9.2). We determined $\xi$ for periods of different instrumental setups and separately for night (incoming shortwave radiation $< 20$ $Wm^{-2}$) and day (incoming shortwave radiation $\geq 20$ $Wm^{-2}$) conditions using the following equation

$$F_c = F_{c,WPL} + \xi \frac{(T_s - T_a)\rho_c}{r_a T_a} \tag{2}$$

where $F_c$ ($kg\,m^{-2}\,s^{-1}$) is the true $CO_2$ flux, $F_{c,WPL}$ ($kg\,m^{-2}\,s^{-1}$) is the WPL-corrected OP $CO_2$ flux, $T_s$ (K) is the instrument surface temperature, $T_a$ (K) the ambient air temperature, $r_a$ ($s\,m^{-1}$) the aerodynamic resistance and $\rho_c$ ($kg\,m^{-3}$) the ambient $CO_2$ density. Prior to $\xi$ optimization, we estimated the instrument surface temperature $T_s$ following the parameterization of Järvi et al. (2009) also separately for nighttime and daytime

$$T_{s,day} = 0.93(T_a - T_0) + 3.17 + T_0 \quad and \quad T_{s,night} = 1.05(T_a - T_0) + 1.52 + T_0 \tag{3}$$

with $T_{s,day}$ (K) and $T_{s,night}$ (K) as instrument surface temperature estimates and $T_0$ set to 273.15 K. We determined the scaling factor as a parameter of equation (2) being the modified Burba et al. (2006) approach from Kittler et al. (2017). For

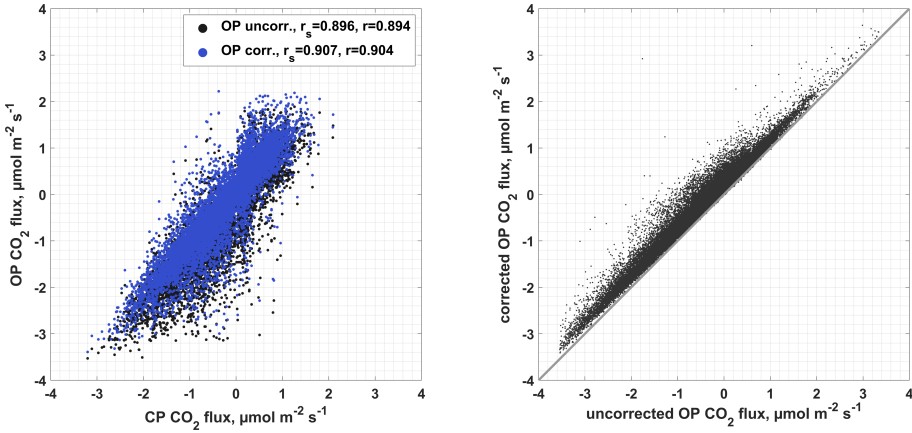

**Figure 3.** Effect of the self-heating correction on the correlation between open-path (OP) and closed-path (CP) fluxes (left panel). Correlations were quantified using Spearman's rank correlation coefficient $r_s$ and Pearson's correlation coefficient $r$. Only quality class 0 is shown. Negative fluxes are affected more strongly by the correction than positive fluxes (right panel).

function fitting, we assumed CP fluxes of quality classes 0 and 1 as true fluxes. We used WPL-corrected OP quality class 0 fluxes and the above described surface temperature estimates as independent variables. Before parameter optimization, we quality-screened the Burba et al. (2006) correction term (expression to the right hand side of $\xi$ in equation (2)) and removed spikes ranging within the uppermost or lowest percent of its distribution. Throughout all years, $\xi$ is larger at daytime than at
nighttime but generally small, adding mostly below 1 % of the full correction term to the uncorrected flux (see Table 4). In four of the seven available years with simultaneous CP and OP fluxes, nighttime $\xi$ optimization converged to values below zero. Before applying the correction models to these periods, we set nighttime $\xi$ estimates to the median of the years yielding parameter values that, including their 95 % confidence bounds, ranged above zero. We used this value and the median of all daytime model optimizations to calculate corrected OP fluxes at times without parallel CP measurements. We did not correct
OP fluxes when radiation measurements or correction term estimates were not available. Correlation between CP and OP fluxes improved throughout all quality classes by applying the self-heating correction (see Table 5), while fluxes indicating net $CO_2$ uptake were affected more strongly than fluxes above zero (see Figure 3).

### 3.5 Carbon dioxide flux gap filling

We used the CP and the corrected OP fluxes (see Figure 4) to compile a $CO_2$ flux time series. We aimed at keeping as
many measured data points as possible, while omitting records with large uncertainty. We accepted all CP values of quality classes 0 and 1. At time steps where no CP fluxes were available, we selected OP values of the same quality classes. The resulting time series contains 75,921 datapoints. Additionally, we filled the remaining gaps in the time series using the marginal distribution sampling (MDS) method as first presented by Reichstein et al. (2005). This method employs two types of model value calculations. The environmental variables global radiation, air temperature and water vapour pressure deficit are binned

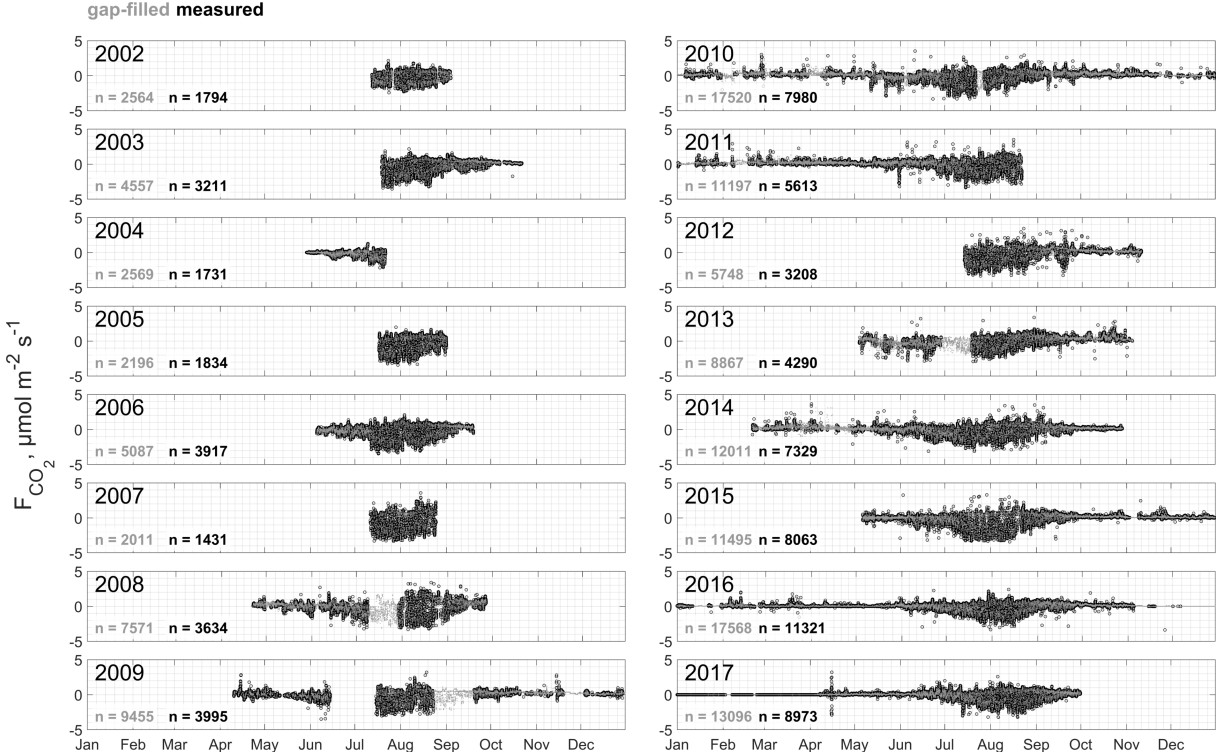

gap-filled **measured**

$F_{CO_2}$, µmol m$^{-2}$ s$^{-1}$

**Figure 4.** Multiannual carbon dioxide flux time series compiled from fluxes measured with closed-path and open-path sensors on Samoylov Island's river terrace. Fluxes of quality class 2 are not shown. Self-heating errors in the OP dataset have been corrected for. Additionally, the result from gap filling this time series with the MDS method is shown. The given number of values for the gap-filled time series include measured fluxes.

in classes and combined in a look-up table (LUT). In case of a gap, flux values related to similar environmental conditions can be looked up and used for averaging and gap filling. The setup of different LUTs for fixed time periods has been first described by Falge et al. (2001). This process can be refined by the use of moving time windows (Moffat et al., 2007) around gaps, as applied by Reichstein et al. (2005). The second model type implemented in the MDS algorithm exploits the commonly high autocorrelation of gas flux time series. The mean diurnal variation (MDV) technique has as well been first described by Falge et al. (2001) and uses the average of available gas flux measurements from adjacent days at the same hour of day to fill a flux gap. The MDS method found wide application, as it has for example been the standard technique within the processing pipeline of the FLUXNET2015 data set, which includes over 1500 site-years of data. The algorithm of Reichstein et al. (2005) combines a screening procedure of the available data for similar environmental conditions (look-up table steps) and the use of a MDV method (diurnal cycle steps) if a gap could not be filled within the look-up table steps. Both techniques include moving windows with variable sizes that are increased until a solution can be found. Large gaps are skipped. To run the gap filling algorithm, we used the REddyProc routine that is accessible through a web-based service hosted by the Department of

Biogeochemical Integration at Max Planck Institute Jena. The R-routine that is executed on this server is a further-developed and extended version of the Reichstein et al. (2005) approach and is described by Wutzler et al. (2017). We did not use the friction velocity filter or the flux partitioning capabilities of the REddy Proc online tool. Gap filling resulted in 131,908 data points. The provided data set includes quality flags for each gap-filled value that depend on the used method and time window size, as defined by Reichstein et al. (2005). These flags take values between 0 and 3, with 0 denoting measurement data, 1 indicating most reliable and 3 least reliable gap-filled fluxes. To assess the overall quality of the gap filling result, the MDS algorithm, in a stepwise manner, treats single available values as gaps and fills them according to the described scheme. Pearson's correlation coefficient between our compiled $CO_2$ flux time series and the MDS quality assessment run, where these values were treated as artificial gaps, is 0.92, with a root mean squared error of 0.31 $\mu$mol m$^{-2}$ s$^{-1}$.

## 3.6 Flux uncertainty estimation

Flux uncertainty can be regarded as a combination of a systematic and a random part. While the attempt should be made to remove systematic biases, random errors cannot be corrected for (Richardson et al., 2012). However, statistical methods exist to estimate the uncertainty of a flux measurement due to random errors. We used three different approaches from literature to quantify random uncertainty and addressed fluxes with a suspected large bias by correcting for it during processing or by filtering in the course of quality assessment.

Most importantly, systematic errors are introduced when underlying EC assumptions are not met. Using the method of Mauder and Foken (2004) that combines an assessment of well developed turbulence and steady state conditions, we identified biased fluxes and flagged them. Other sources of systematic errors that we addressed include for example the angle of attack correction of faulty sonic anemometer readings, filtering for low instrument signal strength, the OP self-heating correction and compensations for high frequency loss and air density fluctuations (see sections 3.2.2, 3.3 and 3.4). Although we are confident that we applied corrections for systematic errors both rigorously and carefully enough, biases were certainly not always removed efficiently. The quality flags included in the data set, reflect a level of confidence based on the assessment of general EC assumptions and our six additional quality filtering steps (see section 3.3).

To be able to include a random uncertainty estimate for each individual OP and CP flux in the provided data set, we set EddyPro to calculate random uncertainty estimates following Finkelstein and Sims (2001). The authors developed a method that aims at quantifying flux uncertainty associated with turbulence sampling errors. These errors can contribute largely to the total random error as they refer to the insufficient sampling of large eddies with high spectral energy. Due to the stochastic nature of turbulence, this type of error is random. To estimate its magnitude, the so-called integral turbulence time scale (ITS) is first determined by expressing the covariance of vertical wind velocity and gas concentration as a function of a lag time between these two time series. The ITS is then given by integrating the cross-correlation function theoretically from 0 to infinity, in practice, however, until an upper lag time limit is reached. The upper limit can be defined in three different ways in EddyPro. We used the definition of the normalized cross-correlation function reaching a value of $1/e = 0.369$ to determine an upper lag time limit used for integration. While the normalized cross-correlation should reach zero with increasing lag time in theory, in practice it sometimes does not. The setting we used on the one hand provides the least conservative estimate of

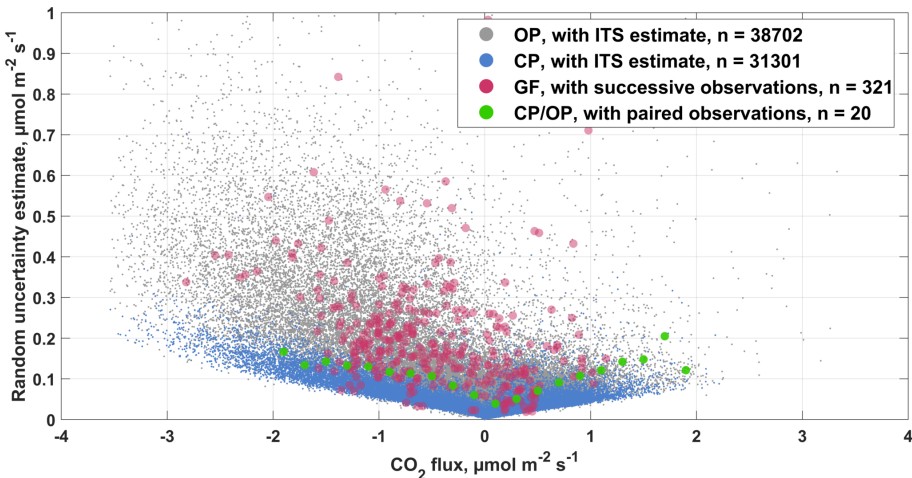

**Figure 5.** Random uncertainty estimates for all closed path (CP) and open-path (OP) $CO_2$ fluxes calculated using (1) estimates of the integral turbulence time scale (ITS), (2) the successive observations approach and results from gap filling (GF) and (3) the paired observations approach during periods with simultaneous OP and CP records.

the ITS but on the other hand offers computational efficiency and makes sure that an upper limit for integration can reliably be found. With the ITS, a flux uncertainty can be determined by calculating the variance of an EC flux or, as Finkelstein and Sims (2001) put it, by calculating the variance of the covariance. This ensemble variance would approach zero with the averaging time approaching infinity. In the data set available for download, a random uncertainty estimate calculated with the method of Finkelstein and Sims (2001) is given for each OP and CP flux (see Table 7). Random uncertainties based on ITS estimation

observations increase with absolute fluxes with mean values of 0.16 and 0.05 $\mu mol\,m^{-2}\,s^{-1}$ for OP and CP fluxes (see Figure 5). OP random uncertainty estimates are generally larger and more scattered with respect to the corresponding flux values.

As the above described random uncertainty estimate specifically addresses the turbulence sampling error, other sources of random flux errors such as the noise introduced by the different components of the measurement system are neglected. With simultaneous measurements from two sensors, we could additionally estimate random errors for the measurement system as

a whole during times when the data sets from both sensors overlapped. We followed the paired observations approach as presented by Dragoni et al. (2007) and calculated a random error estimate $\epsilon$ as

$$\epsilon = \frac{1}{\sqrt{2}} \cdot (F_{CP} - F_{OP}) \tag{4}$$

with the closed-path and open-path $CO_2$ fluxes $F_{CP}$ and $F_{OP}$ of quality classes 0 and 1 in $\mu mol\,m^{-2}\,s^{-1}$. The distribution of $\epsilon$ estimates is shown in Figure 6. The $\epsilon$ values calculated with OP fluxes corrected for the self-heating error have a mean

close to zero and are distributed more symmetrically than the $\epsilon$ values calculated with uncorrected OP fluxes. The mean of this distribution is shifted from its mode as well as from zero, indicating a much stronger systematic component whithin the measurement error. This result increases our confidence that the OP self-heating correction we applied was successful in removing a systematic bias from the data. Further following Dragoni et al. (2007), we used the $\epsilon$ system error data set from

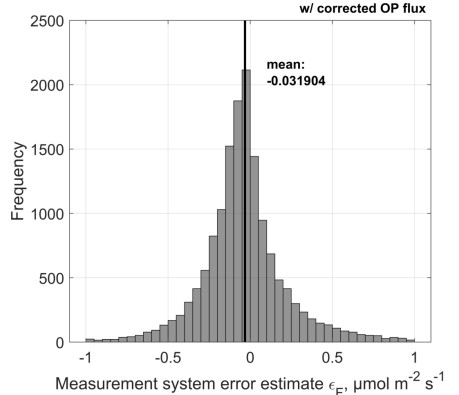
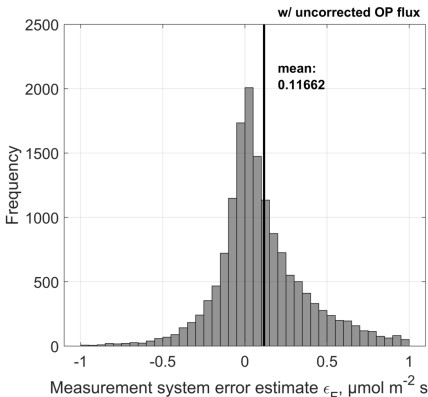

**Figure 6.** Distributions of the measurement system errors $\epsilon$ estimated using the paired observations approch for differences between closed path and corrected (left panel) as well as uncorrected (right panel) open-path (OP) fluxes.

the overlap period to generate flux uncertainty estimates for bins of increasing OP flux ranges. We sorted the $\epsilon$ values in 20 corresponding flux bins between -2 and 2 $\mu\mathrm{mol\,m^{-2}\,s^{-1}}$ and calculated an uncertainty estimate for each bin $\sigma(\epsilon)_i$ as

$$\sigma(\epsilon)_i = \sqrt{2}\frac{1}{N_j}\sum_{j=0}^{N_j}|\epsilon_{i,j} - \overline{\epsilon}_i| \tag{5}$$

Results show (see Figure 5) a similar data range and pattern of uncertainty estimates in relation to associated fluxes like the half-hourly values calculated after Finkelstein and Sims (2001).

As a third method of random uncertainty estimation we simplified the successive observations approach from Richardson et al. (2006) by using results of the quality run performed during MDS gap filling (see section 3.5). We selected the time steps when an flux observation and a MDS value that was estimated using a one day window and the MDV technique were available. We used the standard deviation of the fluxes measured at the same hour of day within a one day window, as an uncertainty estimate of the observed flux. Results are shown in Figure 5 and also increase with rising absolute fluxes in the same ranges as
random uncertainties due to turbulence sampling error or measurement system error do.

We included the results obtained with ITS estimation into the uploaded data set considering the similarity between the uncertainty–flux relations calculated with independent methods as well as due to the advantage of a distinct uncertainty estimate for each sensor and time step.

### 3.7  Footprint modeling

In order to quantify the cumulative contribution of distinct surface classes to the EC source area, we evaluated the two-dimensional analytical footprint formulation described by Kormann and Meixner (2001) in combination with a $0.14\,\mathrm{m} \times 0.14\,\mathrm{m}$ resolution surface classification of Samoylov Island's central river terrace provided by Muster et al. (2012). The authors divide the surface into four classes based on hydrology and vegetation communities, as illustrated in Figure 2. Kormann and Meixner (2001) presented an analytical solution to the crosswind-distributed advection-diffusion equation described by Van Ulden

(1978) and Horst and Weil (1992). Using the analytical model of Huang (1979), the authors solved the power-law profiles of horizontal wind speed and eddy diffusivity by relating them to the Monin-Obukhov similarity theory, including the stability dependence of the exponents in the power laws at a certain height. We implemented the equations given in Kormann and Meixner (2001) as a Matlab (v. 9.2) function and added a quality filter, omitting calculations when friction velocity was larger than $0.9\,\mathrm{m\,s^{-1}}$ or smaller than $0\,\mathrm{m\,s^{-1}}$, wind speed was below zero or above $20\,\mathrm{m\,s^{-1}}$, the crosswind standard deviation was

below zero or above $3\,\mathrm{m\,s^{-1}}$ or Monin-Obukhov length was smaller than $10^{-3}$ m or larger than $10^4$ m. Prior to half-hourly footprint calculations, we additionally determined roughness length statistics for annual subsets of data and binned them in 2 ° wind direction classes. The medians of these classes were used in the subsequent half-hourly footprint estimation, depending on the mean wind direction during these 30 minutes. We evaluated the footprint model at the same resolution that was used by (Muster et al., 2012) to classify the surface (i. e. 0.14 m × 0.14 m). We could thereafter assign a probability of being the EC

source area to each classified pixel and sum up the probabilities of all pixels belonging to the same surface class to estimate the contribution of each class. This proceeding to combine an EC source area estimation with a land cover classification is similar to what has been applied and described in more detail by Forbrich et al. (2011).

## 4 Discussion

Although we did our best to ensure the consistency and appropriateness of the data processing workflow for the presented NEE

time series, due to technical and logistical constraints during 16 years of field work, disparities in the experimental setup exist which may challenge its integrity. The EC tower was relocated twice, the measurement height was changed three times (see Figure 2 and Table 1). These changes of tower location and measurement height affected the source area and hence the surface types sampled during flux measurements. Most notably, between July 2007 and June 2009, the EC tower was placed about 650 m south-west of its original position at the center of Samoylov Island, in an area with an increased coverage of the surface class

*wet tundra*. This is revealed by the footprint analysis (Figure 6). While the EC footprint is dominated by the surface class *dry tundra* throughout the time series, during subperiods 2007, 2008 and 2009 I the contributions of *wet tundra* to the measured flux are significantly higher. To check the effect of the shifts in tower location and measurement height on cumulative $CO_2$-C fluxes, we calculated flux sums for a period when flux time series without gaps were available in most years. The overlapping period covers days of year 200 to 234, i.e. part of the growing season in all years except for 2004 (see Figure 8). Interannual

variability of cumulative C fluxes in years with constant tower location (and measurement height) appears to be large and driven by a more complex set of variables than shifts in surface class contributions only. Flux sums from the periods when EC tower relocation led to a significant shift in EC footprint composition are well within the range of the distribution of cumulated fluxes from years with a more homogeneous EC fetch area. We therefore assume that, at least with respect to budget calculations, the presented long-term time series is not disrupted and can be regarded as representative for a polygonal tundra site dominated

by *dry tundra*. For a more in depth analysis of flux dynamics, footprint information should and can be considered by users of the data set. Recently, a comparison between surface class level NEE models based on chamber measurements with EC fluxes, using the half-hourly footprint information provided in this data set for scaling, yielded good agreement between the

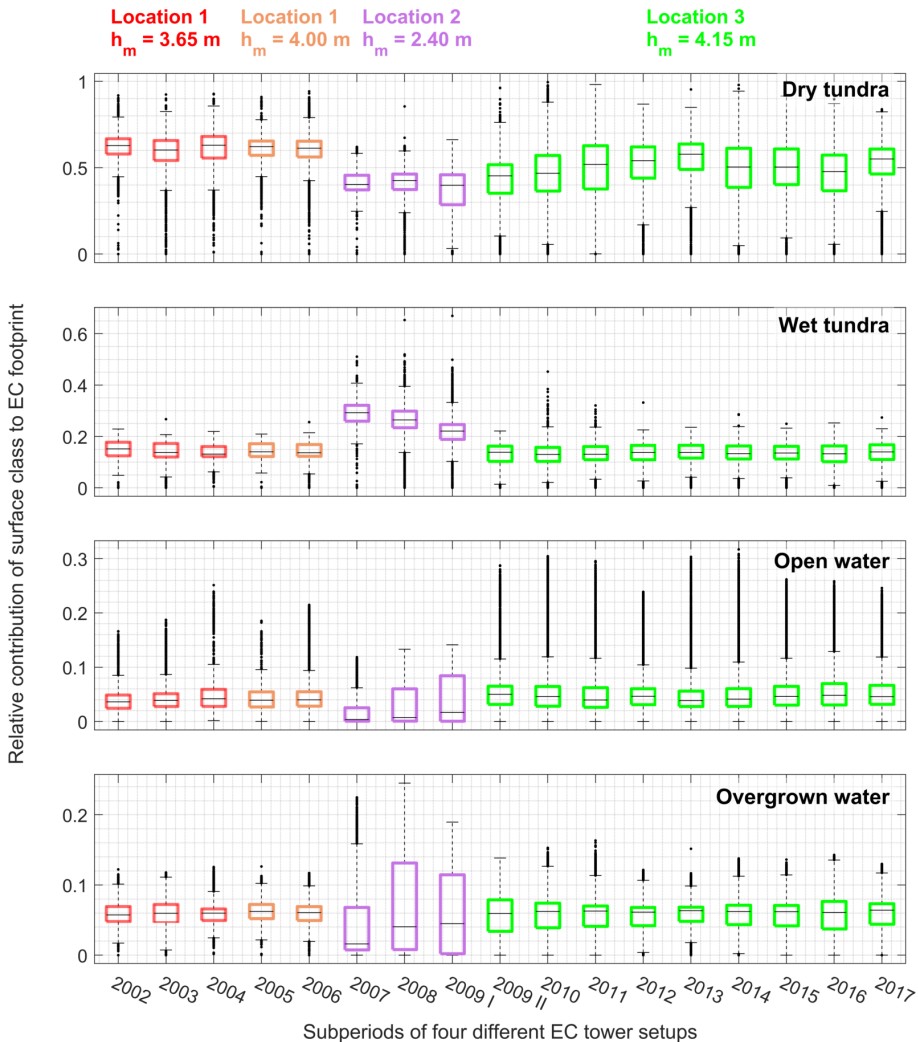

**Figure 7.** Mean surface class composition of the eddy covariance footprint during 17 subperiods of four different tower setups at three locations on Samoylov Island.

results obtained with both methods (Eckhardt et al., 2018). We regard the availability of half-hourly footprint information in the presented NEE data set an attribute that sets it apart from other studies and holds chances for comprehensive analyses.

Apart from the changes in anemometer height, other deviations of the general instrument setup occurred due to limitations in data storage during two winter periods when the acquisition frequency was reduced to 5 Hz and 10 Hz respectively. Rinne et al. (2008) demonstrated in a field experiment that fluxes calculated from raw data recorded at frequencies below 20 Hz compare well with fluxes derived from high frequency raw data. Differences arise as an increase of random noise and not as a systematic bias. High frequency noise removal before ensemble spectra estimation in EddyPro is effective in limiting the effect of increased noise on the quality of transfer function estimation in the process of spectral correction. Overall spectral

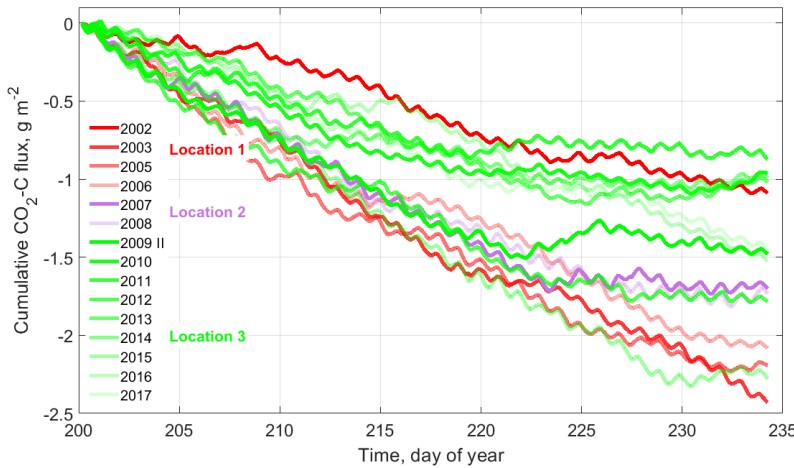

**Figure 8.** Comparison of cumulative $CO_2$ flux sums of different years during the same day of year range.

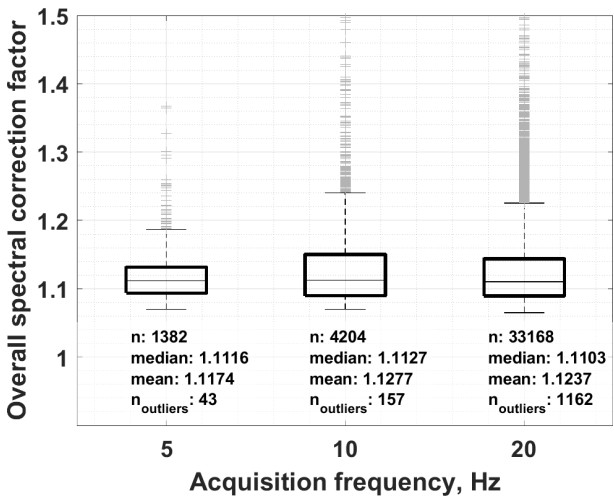

**Figure 9.** Spectral correction factor statistics for periods with different acquisition frequencies.

correction in EddyPro is expressed as a spectral correction factor (SCF) which comprises the effect of all applied compensations for high and low frequency loss. Raw fluxes are multiplied with the respective SCFs during processing. We compared the SCF distributions of the two above mentioned winter periods with statistics of the remaining parts of the time series when data was recorded at 20 Hz. SCF deviations between the different acquisition frequencies are minor (see Figure 9) what implies that systematic differences between fluxes calculated from raw data of different temporal resolutions are in fact small; random uncertainties increase, however.

## 5  Scientific overview

While results on methane exchange fluxes and the soils' methane production and oxidation potential are more prominent in the publication record (e.g. Wagner et al., 2003; Kutzbach et al., 2004; Liebner and Wagner, 2007; Knoblauch et al., 2008; Sachs et al., 2008; Wille et al., 2008; Schneider et al., 2009; Sachs et al., 2010; Liebner et al., 2011; Knoblauch et al., 2015),

literature on $CO_2$ flux time series recorded with the same measurement system presented in this publication is available for distinct years. Flux processing has, however, been streamlined only now. The length of the time series, the addition of detailed footprint information, the site-specific correction of OP fluxes and the coherent processing and quality filtering distinguishes the data set at hand from past publications like the contribution made to the FLUXNET2015 data set (Kutzbach et al., 2015).

Ongoing analysis of the long-term data set (Kutzbach, unpublished) *inter alia* confirms what has been found in the past

(Kutzbach, 2006; Kutzbach et al., 2007; Runkle et al., 2013). The polygonal tundra of Samoylov Island appears to be a robust growing season $CO_2$-C sink whereas this sink strength can vary that much interannually that prolonged low-level respiratory $CO_2$-C loss during the cold season can offset $CO_2$-C uptake during the vegetation period. Reduced summer uptake has been observed for both the coldest and warmest summers. Runkle et al. (2013) found that with frequent early season heat spells, the temperature-induced increase in respiratory release can exceed the rise in photosynthetic uptake. Recently, all data from

this publication has been contributed to the Arctic Data Center's chamber and EC synthesis project *Reconciling historical and contemporary trends in terrestrial carbon exchange of the northern permafrost-zone* that aims at identifying seasonal and interannual C flux dynamics and its drivers based on a newly established pan-arctic data base.

In context with the improvement of earth system models (ESMs), carbon dioxide fluxes from Samylov Island can be especially of use due to the site's comparably high moss cover. Using data from Samoylov, Chadburn et al. (2017) found that current

ESMs miss an observed early season $CO_2$ uptake peak suspected to be connected to the earlier onset of moss photosynthesis in comparison with vascular plants. Although there have been advances and e. g. Porada et al. (2013) developed a dynamic moss model for JSBACH (Raddatz et al., 2007), Chadburn et al. (2017) noted that the simulated $CO_2$ uptake and release terms combining vascular vegetation and moss carbon fluxes did not agree with observational data. The fact that the Samoylov Island NEE data set has now been extended and its quality has been greatly improved holds the opportunity to estimate the

performance of updated ESM versions that are set up to represent carbon fluxes in the moss layer better.

## 6  Data availability

The data set was uploaded to the Pangaea data base (Holl and Kutzbach, 2018) and can be accessed through https://doi.pangaea. de/10.1594/PANGAEA.892751. The included columns are given in Table 7. Ancillary long-term time series of meteorological and soil variables from Samoylov Island are available from Boike et al. (2018) and can be accessed through https://doi.pangaea.

de/10.1594/PANGAEA.891142.

# 7 Conclusions

We are confident that the presented carbon dioxide land–atmosphere flux data set is of high quality and is likely to be of value to the scientific community. We screened the data carefully and applied filtering rules to identify erroneous data, taking into account sensor diagnostics, time lag statistics and the presence of atmospheric conditions that allow for a robust application of the EC method. We followed standardized processing and quality control/assurance routines to allow for comparability between different years from our site as well as with flux time series from other tundra environments. With OP measurements being paralleled by CP measurements in seven years, we had the opportunity to correct for self-heating errors in our OP measurements with a site-specifically scaled correction term, rather than using default correction methods (e. g. Burba et al., 2008). We could therefore address different sensor setups with different correction terms and thereby improve our OP data set, as the self-heating effect has distinct impacts on sensors installed at different inclinations. We quantified the contribution of certain soil and vegetation community types to each half-hourly EC footprint, taking into account varying roughness lengths throughout different years and wind direction sectors. We estimated the cumulative probability of being the EC source area for the four main surface classes on Samoylov Islands' river terrace by using a classified image and by computing an analytical footprint model. Multiannual results show (see Table 6) that on average the combination of different surface classes within the EC footprint is representative for the surface composition of the whole river terrace that developed as a polygonal tundra landscape. According to Muster et al. (2012) the river terrace is composed of 65 % *dry tundra*, 19 % *wet tundra* and 16 % ponds (sum of *open water* and *overgrown*). On average, the surface class compositions within the EC footprint are very similar to these values. Deviations arise, however, in the years between 2007 and 2009, when the tower location was shifted from the center towards the south-western cliff of Samoylov Island. Nevertheless, the contributions of each surface class to the EC footprint are not only available on average, as presented in Table 6, but half-hourly in the uploaded data set, ensuring that EC source area deviations are quantifiable by a potential user. 16 years of consistently processed and quality-controlled carbon dioxide fluxes from a polygonal tundra landscape typical for Arctic lowlands are a valuable addition to the already existing data base of $CO_2$ net ecosystem exchange observations from the Arctic, especially because of the site's location in Northern Siberia, from where only limited data is available up to now. Furthermore, analysis of this NEE time series is not limited to the gas flux data only. An extensive data stream of meteorological and soil variables between 2002 and 2017 has recently been published by Boike et al. (2018). The authors made their records publicly accessible on the two long-term repositories Pangaea (https://doi.pangaea.de/10.1594/PANGAEA.891142) and Zenodo (https://zenodo.org/record/2223709). The fact of parallelly available ancillary ecosystem variables enables a potential user to put the gas flux dynamics reported in this publication into context with the variability of other ecosystem properties and potential flux drivers. We regard this type of analysis as vital to understand inter-annual variability of gas fluxes and are working on it ourselves (Kutzbach, unpublished).

*Competing interests.* No competing interests are present.

*Acknowledgements.* Without the dedicated work of many scientists, logistics experts and engineers over the years, we would not have been able to present this long-term eddy covariance NEE data set. We want to thank Niko Bornemann, Tim Eckhardt, Mauel Helbig, Lars Heling, Oliver Kaufmann, Zoé Rehder, Norman Rößger, Norman Rüggen, Günter Stoof and Waldemar Schneider for their commitment, diligence and ingenuity. We thank Jakob Sievers for providing us with a starting point for the Matlab implementation of the Kormann and Meixner (2001) footprint model and Norman Rößger for sharing his analysis of the long-term meteorological data from Tiksi with us. This work was supported through the Cluster of Excellence CliSAP (EXC177), Universität Hamburg, funded through the German Science Foundation (DFG), by the European Commission through the project PAGE21 (FP7-ENV-2011, 282700) and by the German Ministry of Education and Research (BMBF) through the projects CarboPerm (03G0836A) and KoPf (03F0764A).

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

**Table 1.** List of deployed instrument types. All infrared gas analyzers were manufactured by LI-COR Biosciences (USA), R3 sonic anemometers were built by Gill Instruments Ltd. (UK), CSAT3 anomemeters by Campbell Scientific Ltd. (UK)

| | Gas analyzer | | Anemometer | | Data coverage | |
| Year | Closed path | Open path | Model | Height, m | Date range | Days |
|---|---|---|---|---|---|---|
| 2002 | LI-7000 | n/a | R3 | 3.65 | 12-Jul to 03-Sep | 53 |
| 2003 | LI-7000 | n/a | R3 | 3.65 | 19-Jul to 22-Oct | 95 |
| 2004 | LI-7000 | n/a | R3 | 3.65 | 28-May to 20-Jul | 53 |
| 2005 | LI-7000 | n/a | R3 | 4 | 17-Jul to 01-Sep | 46 |
| 2006 | LI-7000 | n/a | R3 | 4 | 05-Jun to 19-Sep | 106 |
| 2007 | n/a | LI-7500 | CSAT3 | 2.4 | 11-Jul to 23-Aug | 36 |
| 2008 | n/a | LI-7500 | CSAT3 | 2.4 | 22-Apr to 26-Sep | 157 |
| 2009 I | n/a | LI-7500 | CSAT3 | 2.4 | 10-Apr to 14-Jun | 65 |
| 2009 II | n/a | LI-7500 | CSAT3 | 4.15 | 15-Jul to 29-Dec | 167 |
| 2010 | LI-7000 | LI-7500 | CSAT3 | 4.15 | 01-Jan to 31-Dec | 359 |
| 2011 | LI-7000 | LI-7500 | CSAT3 | 4.15 | 01-Jan to 22-Aug | 233 |
| 2012 | n/a | LI-7500 | CSAT3 | 4.15 | 13-Jul to 10-Nov | 120 |
| 2013 | LI-7000 | LI-7500A | CSAT3 | 4.15 | 04-May to 05-Nov | 185 |
| 2014 | LI-7000 | LI-7500A | CSAT3 | 4.15 | 21-Feb to 29-Oct | 250 |
| 2015 | LI-7000 | LI-7500A | CSAT3 | 4.15 | 06-May to 31-Dec | 239 |
| 2016 | LI-7000 | LI-7500A | CSAT3 | 4.15 | 01-Jan to 19-Nov | 323 |
| 2017 | LI-7000 | LI-7500A | CSAT3 | 4.15 | 01-Jan to 30-Sep | 272 |

**Table 2.** Eddy covariance flux processing steps. Partly differing processing was applied to raw data from closed and open-path analyzers. OP and CP fluxes were computed consistently for the whole period from 2002 to 2017. Setup-dependent statistics (for time lags and *in situ* spectral correction methods) were evaluated annually or if tower position, CP pump speed or any other analyzer metadata changed.

| Processing step | Method | |
|---|---|---|
| | **Closed path data** | **Open path data** |
| **Spike detection & removal** | raw data spike removal (Vickers and Mahrt, 1997) | |
| **Angle of attack correction** | from 2002 to 2006 during Gill anemometer deployment (Nakai et al., 2006) | n/a, sensor was not deployed between 2002 and 2006 |
| **Axis rotation** | Double rotation (Kaimal and Finnigan, 1994) | |
| **Detrending** | linear, (Gash and Culf, 1996) | |
| **Correction for air density fluctuations** | sample-wise conversion of raw data to mixing ratios (Ibrom et al., 2007b; Burba et al., 2012) | application of WPL-Terms to fluxes (Webb et al., 1980) |
| **Time lag compensation** | covariance maximization with nominal time lag from statistics | covariance maximization |
| **Spectral corrections for** | | |
| High-pass filtering | analytic (Moncrieff et al., 2004) | |
| Low-pass filtering | *in situ*/analytic (Ibrom et al., 2007a) | analytic (Moncrieff et al., 1997) |
| Instrument separation | Horst and Lenschow (2009) | n/a |
| **Eddy Pro version** | $\geq 6.0.0$ | |

**Table 3.** Additional quality flagging steps after flux processing. Flagged fluxes were assigned to quality class 2 if not in this class already according to the Mauder and Foken (2004) quality assessment. As CP time lag detection quality had been addressed earlier during flux processing in EddyPro, it was not screened at this stage.

| Step | Applied to | | # of flagged fluxes | |
|---|---|---|---|---|
| | OP fluxes | CP fluxes | OP | CP |
| 1: Raw data skewness/kurtosis | yes | yes | 23769 (23 %) | 12043 (18 %) |
| 2: Instrument signal stregth | yes | no | 6951 (7 %) | n/a |
| 3: Time lag detection quality | yes | no | 20277 (20 %) | n/a |
| 4: Abolute concentration limits | yes | yes | 223 (0.2 %) | 2261 (3 %) |
| 5: Exclusion of outliers when simultaneous CP fluxes close to zero | yes | n/a | 346 (0.3 %) | n/a |
| 6: Absolute flux limits | yes | yes | 634 (0.6 %) | 102 (0.6 %) |

**Table 4.** Estimates of scaling factor $\xi \pm 95\%$ confidence intervals used for open-path flux correction. $\xi$ describes the portion of the self-heating correction term, given by Burba et al. (2006) for vertically installed instruments, that is needed to correct OP fluxes determined with inclined gas analyzers. The scaling factor was optimized as a parameter of a nonlinear function where CP data were regarded as true fluxes. It was therefore determined for years when parallel CP and OP measurements were available. In case of an optimization converging to unreasonable values (below zero), we used the median of the remaining $\xi$ estimates.

| Year | Daytime $\xi$ | Nighttime $\xi$ |
|------|---------------|------------------|
| **2010** | $0.0076 \pm 0.0012$ | $0.0071 \pm 0.0013$ |
| **2011** | $0.0116 \pm 0.0009$ | $0.0068 \pm 0.0015$ |
| **2013** | $0.0150 \pm 0.0007$ | $0.0104 \pm 0.0009$ |
| **2014** | $0.0094 \pm 0.0006$ | $0.0071$ |
| **2015** | $0.0050 \pm 0.0010$ | $0.0071$ |
| **2016** | $0.0051 \pm 0.0005$ | $0.0071$ |
| **2017** | $0.0069 \pm 0.0005$ | $0.0071$ |

**Table 5.** Spearman's rank correlation coefficient $r_s$ and Pearson's correlation coefficient $r$ between closed-path (CP) and open-path (OP) fluxes with and without the applied self-heating correction. The agreement between CP and OP fluxes increases throughout all quality classes after OP correction.

|       |               | Quality class 0 | Quality classes 0,1 | Quality classes 0, 1, 2 |
|-------|---------------|-----------------|---------------------|-------------------------|
| $r_s$ | OP uncorrected | 0.896           | 0.866               | 0.508                   |
|       | OP corrected   | 0.907           | 0.871               | 0.512                   |
| $r$   | OP uncorrected | 0.894           | 0.871               | 0.042                   |
|       | OP corrected   | 0.904           | 0.877               | 0.055                   |

**Table 6.** Normalized mean contributions of the surface classes defined by Muster et al. (2012) to the eddy covariance footprint. Values were averaged over each subperiod and normalized to sum up to 1. Additionally, the mean non-normalized sum of all surface class contributions is given as column *Image contribution*. These values indicate how sufficient the classified area is to describe the EC footprint. Non-normalized half-hourly contributions of the single classes are given in the provided data set.

| Year | Tundra | | Water | | Median image |
| --- | --- | --- | --- | --- | --- |
| | *dry* | *wet* | *overgrown* | *open* | contribution |
| **2002** | 0.71 | 0.17 | 0.07 | 0.05 | 0.88 |
| **2003** | 0.70 | 0.17 | 0.07 | 0.05 | 0.87 |
| **2004** | 0.71 | 0.16 | 0.07 | 0.06 | 0.88 |
| **2005** | 0.71 | 0.17 | 0.07 | 0.05 | 0.87 |
| **2006** | 0.70 | 0.17 | 0.07 | 0.06 | 0.86 |
| **2007** | 0.54 | 0.37 | 0.06 | 0.02 | 0.73 |
| **2008** | 0.53 | 0.34 | 0.09 | 0.04 | 0.77 |
| **2009 I** | 0.54 | 0.32 | 0.08 | 0.06 | 0.72 |
| **2009 II** | 0.64 | 0.19 | 0.09 | 0.08 | 0.71 |
| **2010** | 0.65 | 0.18 | 0.09 | 0.08 | 0.73 |
| **2011** | 0.67 | 0.18 | 0.08 | 0.07 | 0.79 |
| **2012** | 0.67 | 0.18 | 0.08 | 0.07 | 0.80 |
| **2013** | 0.69 | 0.17 | 0.08 | 0.06 | 0.83 |
| **2014** | 0.66 | 0.18 | 0.08 | 0.07 | 0.77 |
| **2015** | 0.66 | 0.18 | 0.08 | 0.08 | 0.78 |
| **2016** | 0.65 | 0.18 | 0.09 | 0.08 | 0.74 |
| **2017** | 0.67 | 0.18 | 0.08 | 0.07 | 0.82 |

**Table 7.** Description of columns included in the data set file.

| Column name | Unit/Format | Description |
| --- | --- | --- |
| Date/Time (Local) | yyyy-mm-ddTHH:MM | Timestamp referring to end of 30 minute flux calculation period in local time (UTC+9h). |
| Date/Time (UTC) | yyyy-mm-ddTHH:MM | Timestamp referring to end of 30 minute flux calculation period in UTC. |
| CP $CO_2$ flux | $\mu mol\, m^{-2}\, s^{-1}$ | Closed path $CO_2$ flux |
| QC CP $CO_2$ flux | dimensionless | Closed path $CO_2$ flux quality classes 0, 1 and 2 |
| CP $CO_2$ flux rand unc | $\mu mol\, m^{-2}\, s^{-1}$ | Closed path $CO_2$ flux random uncertainty estimate (Finkelstein and Sims, 2001) |
| OP $CO_2$ flux | $\mu mol\, m^{-2}\, s^{-1}$ | Open path $CO_2$ flux |
| OP corr $CO_2$ flux | $\mu mol\, m^{-2}\, s^{-1}$ | Corrected open-path $CO_2$ flux (Kittler et al., 2017) |
| QC OP $CO_2$ flux | dimensionless | Open path $CO_2$ flux quality classes 0,1 and 2 |
| OP $CO_2$ flux rand unc | $\mu mol\, m^{-2}\, s^{-1}$ | Open path $CO_2$ flux random uncertainty estimate (Finkelstein and Sims, 2001) |
| $CO_2$ flux comp | $\mu mol\, m^{-2}\, s^{-1}$ | Time series compiled of open and closed-path quality class 0 and 1 fluxes |
| $CO_2$ flux gf | $\mu mol\, m^{-2}\, s^{-1}$ | Gap-filled $CO_2$ flux time series |
| QC $CO_2$ flux gf | dimensionless | Quality flag of gap-filled fluxes, between 0 and 3 (Reichstein et al., 2005) |
| $CO_2$ flux gf std | $\mu mol\, m^{-2}\, s^{-1}$ | Standard deviation of gap-filled flux estimates, calculated from the data used for averaging |
| FP CC dry | dimensionless | Contribution of surface class *dry tundra* to the eddy covariance footprint |
| FP CC wet | dimensionless | Contribution of surface class *wet tundra* to the eddy covariance footprint |
| FP CC ove | dimensionless | Contribution of surface class *overgrown water* to the eddy covariance footprint |
| FP CC wat | dimensionless | Contribution of surface class *open water* to the eddy covariance footprint |