# Peer review of "A long-term (2002 to 2017) record of closed-path and open-path eddy covariance $CO_2$ net ecosystem exchange fluxes from the Siberian Arctic"

_Earth System Science Data, 2018_

## Referee Comment (RC1) · Anonymous Referee #1 · 19 Nov 2018

Review on the manuscript 'A long-term (2002 to 2017) record of closed-path and open-path eddy covariance CO2 net ecosystem exchange fluxes from the Siberian Arctic', submitted for publication to ESSD by David Holl et al.

This manuscript describes a long-term (16 years) record of eddy-covariance measurements of CO2 exchange fluxes between a Northern Siberian tundra ecosystem and the atmosphere. Accompanying a data publication in PANGAEA, the text provides a comprehensive background on site characteristics, data processing steps and quality assurance measures. A specific focus is placed on the use of closed-path versus open-path gas analyzers, and how related systematic errors were corrected to merge

both time series into a single dataset.

Overall, the manuscript is well structured and well written, and includes the major elements that a reader will need to understand the main characteristics of the flux dataset published in PANGAEA. Due to its unique position in Northern Siberia, the Samoylov Island observation site is of high importance for Arctic climate change research. Therefore, this 16-year time series of eddy-covariance fluxes, derived with uniform data processing and quality assessment protocols, is clearly relevant for the Arctic research community.

While I do not have any major concerns with the manuscript in its current form, I strongly recommend a number of 'medium' changes that, while probably straightforward to address, should further strengthen the text:

1.) Inclusion of a 'scientific overview' In the 'Site description' Section, the first 5 paragraphs give a comprehensive overview on the site conditions, while the last paragraph is clearly detached from this material, and in its present form does not belong there. Still, I believe it will be of use to the reader to demonstrate what has been found so far based on the flux time series presented in this manuscript. My recommendation is to move this paragraph to a new chapter 4, i.e. between methods and data availability, and extend it to a length of 3-4 paragraphs in total. This would give ample room to summarize the main findings based on Samoylov eddy-covariance (and other) data so far, therefore highlighting the value of the dataset presented herein, and the role of the site in general for Arctic climate change research.

2.) Ensure that tower locations do not disrupt continuous time series The combination of text in Section 3.1, Figure 1 and Table 1 provides a good overview on the different site setups used to form this 16-year data record. However, the material also raises the question how the shifts in tower position and sensor configuration, including sensor height, may have influenced the signal captured by the EC system, and therefore maybe biased the long-term time series. I therefore recommend moving Section 3.6
upward as a new Section 3.2, and extending the discussion of the footprint issue. You can use parts of the conclusions section for this, but more details need to be provided how the shifts in landscape element fraction in the footprints may have compromised the continuity of the flux observations. See also my comment on Section 3.6 in the 'line comments' below.

3.) Flux uncertainty description, and discussion A clear definition of data uncertainty is mandatory for publications in this journal. In Section 3.2, you briefly mention that you used the standard EddyPro feature to estimate random flux uncertainties – which is a good start, but certainly deserves more attention. So please work out in a separate paragraph what these random uncertainties consist of, and how exactly those were addressed in EddyPro. Moreover, there are also potential sources of systematic uncertainties in eddy covariance flux measurements, e.g. data-processing errors, or instrument calibration issues. These should ideally be covered directly in your uncertainty assessment of the flux data. Since you obviously decided to ignore them here, you should at least provide a convincing rationale why this simplification is justified.

I strongly recommend to consider these comments, as well as the detailed line comments below, to further strengthen this manuscript. Overall, I recommend this text for publication in ESSD after minor revisions.

Line comments: p.1, abstract & introduction: Within these sections, I'm missing data-driven insights. Having a 16-year data record at hand, I would first think about analyzing the data directly to determine long-term trends in surface-atmosphere exchange processes. Next, I would aim at generating process insights, e.g. what causes interannual and inter-seasonal variability in flux rates, Only then I would start thinking about the time series being a useful resource for calibrating and validating process models. I think these data-driven topics deserve additional attention in both sections.

p.1, l.6: FLUXNET is not restricted to CO2 fluxes

p.2, l.7: excessive use of references for a single statement

p.2, l.16: not sure what inversion model have to do with the scope of this paper. They are trained on mixing ratio observations, not fluxes.

p.2, l.30f: this section could use a map to show location of the delta, and the island itself

p.3, l.17: there is no high-centered polygon on the entire island ..??

p.3, l.29ff: climatology information given here is certainly useful, but only based on a $\sim$20 year record from the site itself. It may be helpful to compare to longer-term climate records from the region (e.g., for Tiksi there is data starting in the 1930s).

p.4, l.1f: is there any record of snow depth, and its variability?

p.5, l.6ff: you may add the power consumption as another important difference between CP and OP systems.

p.6, l.4ff: even though you spend a few sentences to describe the WPL-approach, you fail to mention that this is about accounting for the influence of density fluctuations

p.7, Section 3.3: It's a bit odd that you start describing some elements of quality flagging already in Section 3.2, and continue with this material here, in the main quality section. This should be cleaned up. Also, you fail to reference Table 3 in the text. Moreover, you should improve the structure of this Section. You begin with a too short general overview on additional quality filters, and how they are used in the overall QC flagging scheme. You then close the section with very similar statements. This should be merged to a single introductory paragraph that clearly states that you applied 6 more quality checks, and if any of them indicated problems, the quality flag was set to 2.

p.8, l.14: The choice of 450ppm as the upper concentration limit seems rather narrow. Can you please justify?

p.9, Fig.2: Figure 2 isn't really informative, since it's hard to distinguish between corrected and uncorrected time series in such a cloud of values. Please think about a different format (box plots?), or just leave out the plots, and show the regression statistics instead in a table.

p.10, Section 3.5: I suppose Figs. 3 & 4 should belong to this section. They are not referred to in the text. Moreover, it's not necessary to show Fig.4, since given the minor absolute shifts in fluxes after Burba correction in this case, the differences between figures are not discernible. As an alternative for Fig.4, it may be interesting to show the gap-filled time series, maybe even in cumulative form?

p.11, Section 3.6: while the method applied to calculate footprints is sufficiently detailed, it is not fully clear how footprint results were combined with the land cover map. What's completely missing here is a reference to the findings, a.k.a. a bottom line. As already mentioned in the 'medium comments' above, this is an important piece of information, since (as shown in Table 1) multiple positions with multiple sensor heights were used over the 16 year data record. The authors clearly need to point out that this mixture of setups is still suitable to form a coherent, long-term time series of flux exchange for this site. It's not sufficient to just briefly mention these results in the conclusions. In particular, the results in Table 5 emphasize that the southernmost tower position, used within the years 2007-2009, featured a quite different composition of landscape elements than the northern site position. The authors need to make an effort to convince the readers that these differences did not result in a significant deviation of flux patterns, and therefore would bias the long-term trends.

p.11, Section 4: It's good to list the parameters given in the PANGAEA dataset in a separate table. However, since this dataset is obviously restricted to CO2 fluxes and their QC parameters, it would be good to also list the source for ancillary meteorological information, if available, since those will be necessary to put the flux time series into context.

p.11, Section 5: The major part of this section should be moved upwards, into a revised version of the 'footprint modeling' Section. For the conclusions itself, it should be sufficient to state the value of the long-term record, its representativeness for the polygonal tundra ecosystem, and a comprehensive overview on what has been done to assure the quality of the material.

---

## Referee Comment (RC2) · Anonymous Referee #2 · 29 Nov 2018

Review ESSD-2018-98 Siberian Permafrost

A login barrier on Pangaea prevents me from downloading the actual .tsv files. As it turns out I have a valid Pangaea login but, once in, I still cannot access the data. THIS VIOLATES ESSD POLICIES AND PREVENT ME OR ANY OTHER USER FROM FULL EVALUATION OF THE DATA!! This must be fixed immediately. All my comments below assume a quality effort on the part of the authors but until the data becomes fully and freely accessible, I must withhold approval of this manuscript.

Page 3 line 15: this sentence should refer to eddy covariance "systems" because more than one instrument type was deployed on three different towers at two different locations.

Page 3 line 30: technically, "low evaporation rates" should lead to dryer than expected atmospheric humidities. Perhaps the authors refer here to soil (moisture) conditions or to a combination of low specific atmospheric water vapour contents with lower temperatures that lead to a relatively high relative humidity?

Page 4, soils: much higher resolution soil mapping documented here than in the northern circumpolar soil atlas (Jones et al. 2010) so I understand better details here. But Jones et al. soil atlas, at least for central Siberia, adopts the "Russian Soil Classification System" while this paragraph references US or FAO definitions. Why? Because permafrost carbon estimates (e.g. Hugelius et al. in ESSD 2013, not cited here but used extensively in Koven and Schuur, both of which these authors do cite) depend substantially and in fact influence soil type classifications (e.g. again see Hugelius) in permafrost regions, this paper should at least document consistency with other soil classification systems? I know this is not a description of Samoylov permafrost soils data set, but we should at least know consistency or valid reasons for inconsistencies between soil classifications systems used by flux vs soil carbon communities?

Page 4 line 20 "contributied"

Page 4 line 25 (and following): "wind speed" I think you actually mean wind velocity because, unlike speed, you need both magnitude and direction?

Page 4 line 28: Need to mention here that, for a period of two years, the tower location moved almost 1 km to the west-southwest?

Page 4 line 30: For one year sampling rate went to 10 Hz, and later for a period of roughly 9 months sampling went to 5 Hz. Opportunity to test sampling and influence on spectral properties of flux calculations?

Page 5 line 2: "… the data set contains year-round fluxes in some years …". But, from Figure 3, only 2016 had anything close to full annual coverage (e.g. roughly 10k valid 0.5-hour observations out of a maximum possible of 17.5k). No other year shows anything close to full four-season data coverage. If the authors contend that 2014 and 2010 also provide "year-round" flux data then they have a very low standard/expectation for what constitutes valid year-round performance which they should share with readers.

Page 6 line 4: "These Webb-Pearman-Leuning (WPL, Webb et al., 1980) terms". 'These' in this case refers to terms necessary to calculate density for OP measurements but sentence as written allows confusion. Better to specify temperature, pressure, water vapour content, etc.

Page 6 line 7: "undisturbed heat fluxes". Reviewer may know what you mean by 'undisturbed' and why you need those, but you have not explained clearly to readers.

Page 6 line 7: "WPL terms" By this point authors should have told readers exactly what they mean when they say 'WPL terms'. Jargon creeps in here, as well as assumption that every reader already knows the intricacies of eddy correlation measurements. Not true! Please rewrite the initial sentences of this paragraph in a clearer form and format.

Why does a reader find Figs 1 and 2 introduced at appropriate points within the text but all Tables and Figs 3 and 4 appearing at the end of the manuscript after text and references. Need to fix this now and check it again during proofreading.

The authors' descriptions of data processing, quality filtering, self-heating corrections, temporal gap filling etc. seem appropriate and well-described. However, we find no assessment of the the two-year period of tower relocation. Mentioned in the introduction and again in the conclusions, but completely absent from the data processing and data quality descriptions. If that relocation does not matter, e.g. had no effect on time series or data quality, then readers must question the footprint analysis, as mentioned in the Conclusion! Based on lack of information here, this statement from the conclusion " … ensuring that EC source area deviations are quantifiable by a potential user" seems unsupportable for at least two years? One also wonders about the earlier documentation that sampling frequencies changed (e.g. 20 Hz to 10 Hz to briefly 5 Hz). Did those changes also have no effect (or no utility) on data processing. The authors seem to expect users to ignore these possibly substantial location sampling issues but, having mentioned both changes (good) they then fail to report corrections or consequences (bad).

Fig. 4 not referenced in text? In the (barely viewable) version provided for review, Figure 3 and Figure 4 look identical, even to having identical numbers (n) of samples. The legends for the two figures differ slightly, but the figures themselves differ not at all. Wrong figure in the wrong place? Or, because we find no mention of Figure 4 in the text, one figure wrongly duplicated? Serious error, needs attention.

Analysis and use of this particular $CO_2$ data will require simultaneous access to observation time series from boreholes, river gauges, water sampling, long-term meteorology, e.g. Boike et al. 2018. Very important to connect these two data sets. If Boike et al. emerges successfully from ESSD, then we need a close explicit link described here. Until Boike et al. appear in ESSD or elsewhere, release of this data seems premature at best. If other sources of necessary soil, radiation, micrometeorlogical, etc. data exist, please reference those as well or instead? Tiksi?

Note reference to methane measurements. Do these authors consider that they have now have sufficient information, biogeochemical and ecological, to construct an annual carbon budget? If so, they should at least assure readers of forthcoming analyses. If not, why not? Lack of winter-season measurements? Can only construct a valid annual budget for, e.g., 2016?

What makes this time series interesting? Why not simply download from FLUXNET2015? Presumably this data serves as important piece of the tundra carbon analysis presented by Zono et al.? For this reader, the authors have not made an adequate case about potential importance and utility. Readers might consider it potentially very important but this statement, again from the conclusion - "a valuable addition to the already existing data base of $CO_2$ net ecosystem exchange observations from the Arctic" - seems weak and vague. Do the authors claim to have produced a unique high-quality data set or just another contribution to FLUXNET. If the former, then ESSD seems an appropriate venue. If the latter, why bother? Publish the entire FLUXNET data set instead? Again, this reader favours the former but the authors have not made a strong case.

---

## Short Comment (SC1) · 14 Dec 2018

This short comment refers to the first point made by Anonymous Referee #2 with respect to the encountered access barrier on the Pangaea website. I contacted Pangaea staff, they have removed the access limitations. The dataset can now be downloaded, logging in is not required.

---

## Author Comment (AC1) · 1 Feb 2019

Author reply to Referee comments from **Anonymous Referee # 1** from 19 November 2018 (https://doi.org/10.5194/essd-2018-98-RC1, 2018 ) on:

**A long-term (2002 to 2017) record of closed-path and open-path eddy covariance CO$_2$ net ecosystem exchange fluxes from the Siberian Arctic**
by David Holl et al.

Reviewer comments (RC)
Author comments (AC)
Manuscript changes (MC)
Please note that figure numbers in this document refer to figures in this document and not to the numbering in the original draft unless denoted otherwise.

RC 1
1.) Inclusion of a 'scientific overview' In the 'Site description' Section, the first 5 paragraphs give a comprehensive overview on the site conditions, while the last paragraph is clearly detached from this material, and in its present form does not belong there. Still, I believe it will be of use to the reader to demonstrate what has been found so far based on the flux time series presented in this manuscript. My recommendation is to move this paragraph to a new chapter 4, i.e. between methods and data availability, and extend it to a length of 3-4 paragraphs in total. This would give ample room to summarize the main findings based on Samoylov eddy-covariance (and other) data so far, therefore highlighting the value of the dataset presented herein, and the role of the site in general for Arctic climate change research.

I moved and extended the paragraphs about scientific findings from the "Site Description" section to a new chapter 5 "Scientific overview" before the conclusions as suggested. Please also note my reply to RC 4 below.

While results on methane exchange fluxes and the soils' methane production and oxidation potential are more prominent in the publication record (e.g. Wagner et al., 2003; Kutzbach et al., 2004; Liebner and Wagner, 2007; Knoblauch et al., 2008; Sachs et al., 2008; Wille et al., 2008; Schneider et al., 2009; Sachs et al., 2010; Liebner et al., 2011; Knoblauch et al., 2015), literature on CO$_2$ flux time series recorded with the same measurement system presented in this publication is available for distinct years. Flux processing has, however, been streamlined only now. The length of the time series, the addition of detailed footprint information, the site-specific correction of OP fluxes and the coherent processing and quality filtering distinguishes the data set at hand from past publications like the contribution made to the FLUXNET2015 data set (Kutzbach et al., 2015).

Ongoing analysis of the long-term data set (Kutzbach, unpublished) *inter alia* confirms what has been found in the past (Kutzbach, 2006; Kutzbach et al., 2007; Runkle et al., 2013). The polygonal tundra of Samoylov Island appears to be a robust growing season CO$_2$-C sink whereas this sink strength can vary that much interannually that prolonged low-level respiratory CO$_2$-C loss during the cold season can offset CO$_2$-C uptake during the vegetation period. Reduced summer uptake has been observed for both the coldest and warmest summers. Runkle et al. (2013) found that with frequent early season heat spells, the temperature-induced increase in respiratory release can exceed the rise in photosynthetic uptake. Recently, all data from this publication has been contributed to the Arctic Data Center's chamber and EC synthesis project *Reconciling historical and contemporary trends in terrestrial carbon exchange of the northern permafrost-zone* that aims at identifying seasonal and interannual C flux dynamics and its drivers based on a newly established pan-arctic data base.

In context with the improvement of earth system models (ESMs), carbon dioxide fluxes from Samylov Island can be especially of use due to the site's comparably high moss cover. Using data from Samoylov, Chadburn et al. (2017) found that current ESMs miss an observed early season CO$_2$ uptake peak suspected to be connected to the earlier onset of moss photosynthesis in comparison with vascular plants. Although there have been advances and e. g. Porada et al. (2013) developed a dynamic moss model for JSBACH (Raddatz et al., 2007), Chadburn et al. (2017) noted that the simulated CO$_2$ uptake and release terms combining vascular vegetation and moss carbon fluxes did not agree with observational data. The fact that the Samoylov Island NEE data set has now been extended and its

quality has been greatly improved holds the opportunity to estimate the performance of updated ESM versions that are set up to represent carbon fluxes in the moss layer better.

RC 2

2.) Ensure that tower locations do not disrupt continuous time series The combination of text in Section 3.1, Figure 1 and Table 1 provides a good overview on the different site setups used to form this 16-year data record. However, the material also raises the question how the shifts in tower position and sensor configuration, including sensor height, may have influenced the signal captured by the EC system, and therefore maybe biased the long-term time series. I therefore recommend moving Section 3.6 upward as a new Section 3.2, and extending the discussion of the footprint issue. You can use parts of the conclusions section for this, but more details need to be provided how the shifts in landscape element fraction in the footprints may have compromised the continuity of the flux observations. See also my comment on Section 3.6 in the 'line comments' below.

I added a new "Discussion" section to the manuscript addressing the effects of tower location shifts and other possible disruptions of the time series' coherency.

Although we did our best to ensure the consistency and appropriateness of the data processing workflow for the presented NEE time series, due to technical and logistical constraints during 16 years of field work, disparities in the experimental setup exist which may challenge its integrity. The EC tower was relocated twice, the measurement height was changed three times (see Figure 1 and Table 1 *(in original draft)*). These changes of tower location and measurement height affected the source area and hence the surface types sampled during flux measurements. Most notably, between July 2007 and June 2009, the EC tower was placed about 650 m south-west of its original position at the center of Samoylov Island, in an area with an increased coverage of the surface class wet tundra. This is revealed by the footprint analysis (Figure 1). While the EC footprint is dominated by the surface class *dry tundra* throughout the time series, during subperiods 2007, 2008 and 2009 I the contributions of *wet tundra* to the measured flux are significantly higher.

To check the effect of the shifts in tower location and measurement height on cumulative $CO_2$-C fluxes, we calculated flux sums for a period when flux time series without gaps were available in most years. The overlapping period covers days of year 200 to 234, i.e. part of the growing season in all years except for 2004 (see Figure 2). Interannual variability of cumulative C fluxes in years with constant tower location (and measurement height) appears to be large and driven by a more complex set of variables than shifts in surface class contributions only. Flux sums from the periods when EC tower relocation led to a significant shift in EC footprint composition are well within the range of the distribution of cumulated fluxes from years with a more homogeneous EC fetch area. We therefore assume that, at least with respect to budget calculations, the presented long-term time series is not disrupted and can be regarded as representative for a polygonal tundra site dominated by dry tundra. For a more in depth analysis of flux dynamics, footprint information should and can be considered by users of the data set. Recently, a comparison between surface class level NEE models based on chamber measurements with EC fluxes, using the half-hourly footprint information provided in this data set for scaling, yielded good agreement between the results obtained with both methods Eckhardt et al. (2018). We regard the availability of half-hourly footprint information in the presented NEE data set an attribute that sets it apart from other studies and holds chances for comprehensive analyses.

Apart from the changes in anemometer height, other deviations of the general instrument setup occurred due to limitations in data storage during two winter periods when the acquisition frequency was reduced to 5 Hz and 10 Hz respectively. Rinne et al. (2008) demonstrated in a field experiment that fluxes calculated from raw data recorded at frequencies below 20 Hz compare well with fluxes derived from high frequency raw data. Differences arise as an increase of random noise and not as a systematic bias. High frequency noise removal before ensemble spectra estimation in EddyPro is effective in limiting the effect of increased noise on the quality of transfer function estimation in the process of spectral correction. Overall spectral correction in EddyPro is expressed as a spectral correction factor SCF which comprises the effect of all applied compensations for high and low frequency loss. Raw fluxes are multiplied with the respective SCFs during processing. We compared the SCF distributions of the two above mentioned winter periods with statistics of the remaining parts of the time series when data was recorded at 20 Hz. SCF deviations between the different acquisition frequencies are minor (see Figure 03) implying that systematic differences between fluxes calculated form raw data of different temporal resolutions are in fact small, random uncertainties increase, however.

[Figure]

Fig. 1     Mean surface class composition of the eddy covariance footprint during 17 subperiods of four different tower setups at three locations on Samoylov Island.

[Figure]

Fig. 2    Comparison of cumulative $CO_2$ flux sums of different years during the same day of year range.

[Figure]

Fig. 3    Spectral correction factor statistics for periods with different acquisition frequencies.

RC 3
3.) Flux uncertainty description, and discussion A clear definition of data uncertainty is mandatory for publications in this journal. In Section 3.2, you briefly mention that you used the standard EddyPro feature to estimate random flux uncertainties – which is a good start, but certainly deserves more attention. So please work out in a separate paragraph what these random uncertainties consist of, and how exactly those were addressed in EddyPro. Moreover, there are also potential sources of systematic uncertainties in eddy covariance flux measurements, e.g. data-processing errors, or instrument calibration issues. These should ideally be covered directly in your uncertainty assessment of the flux data. Since you obviously decided to ignore them here, you should at least provide a convincing rationale why this simplification is justified.

I added a new part "Flux uncertainty estimation" to the "Methods" section.

[revised manuscript text omitted]

Line comments:

RC 4

p.1, abstract & introduction: Within these sections, I'm missing data-driven insights. Having a 16-year data record at hand, I would first think about analyzing the data directly to determine long-term trends in surface-atmosphere exchange processes. Next, I would aim at generating process insights, e.g. what causes interannual and inter-seasonal variability in flux rates, Only then I would start thinking about the time series being a useful resource for calibrating and validating process models. I think these data-driven topics deserve additional attention in both sections.

We regard this dataset publication as a starting point for analysis of flux dynamics done by us and other members of the scientific community. We are aiming at publishing those types of results in the future (Kutzbach, unpublished). In this paper, however, we wanted to focus on the methods we used to process the data rather than its interpretation. To our understanding, this proceeding is in line with the "Aims and scope" of ESSD, which is one reason why we selected this journal.

*"Articles in the data section may pertain to the planning, instrumentation, and execution of experiments or collection of data. Any interpretation of data is outside the scope of regular articles. Articles on methods describe nontrivial statistical and other methods employed (e.g. to filter, normalize, or convert raw data to primary published data) as well as nontrivial instrumentation or operational methods. Any comparison to other methods is beyond the scope of regular articles." (https://www.earth-system-science-data.net/about/aims_and_scope.html)*

RC 5

p.1, l.6: FLUXNET is not restricted to CO2 fluxes

True, this is a lapse. I changed "The site is part of the international network of carbon dioxide flux observation stations (FLUXNET, Site ID: Ru-Sam)." to

The site is part of the international network of eddy covariance flux observation stations (FLUXNET, Site ID: Ru-Sam).

RC 6

p.2, l.7: excessive use of references for a single statement

I do not agree. The reference list is meant to express that many authors agree on the importance of permafrost carbon pools in the context of climate change. The references are thought to proof the statement of "wide recognition" of the topic.

RC 7

p.2, l.16: not sure what inversion model have to do with the scope of this paper. They are trained on mixing ratio observations, not fluxes.

Thank you for pointing out this fact. I changed the sentence to:

McGuire et al. (2012) conclude that reducing uncertainties of regional estimates based on observational data relies on high quality ground-based measurements that should be placed strategically, e. g. along hydrological or vegetation gradients.

RC 8

p.2, l.30f: this section could use a map to show location of the delta, and the island itself

I added an overview map (Figure 6 in this document).

[Figure]

Fig. 6    Location of Samoylov Island (center of panel b) in the Lena River Delta (panel a). Map data from: OpenStreetMap contributors, under Open Database License

RC 9

p.3, l.17: there is no high-centered polygon on the entire island ..??

Yes, there are some high-centered polygons on Samoylov. I changed the sentence.

In contrast to the modern floodplain, the river terrace's surface is patterned due to frost-action that formed a wet polygonal tundra landscape consisting of mostly low-centered and some high-centered ice-wedge polygons as well as thermokarst lakes and channels.

RC 10

p.3, l.29ff: climatology information given here is certainly useful, but only based on a  #20 year record from the site itself. It may be helpful to compare to longer-term climate  records from the region (e.g., for Tiksi there is data starting in the 1930s).

I agree, I added longer-term meteorological information from Tiksi.

The closest WMO (World Meteorological Organisation) weather station is located on the continent, around 110 km southeast from Samoylov Island in the city of Tiksi. Between 1936 and 2017 the mean air temperature reported from Tiksi is – 12.74 °C, mean annual precipitation amounts to 304.5 mm (AARI, 2018). While the mean air temperature in Tiksi is very similar to the 20-year mean from Samoylov Island, average annual precipitation appears to be much higher in Tiksi than in the delta region. Boike et al. (2013) explain this divergence with the fact that Tiksi is located at the coast of the Laptev sea and surrounded by mountains.

RC 11

p.4, l.1f: is there any record of snow depth, and its variability?

I added information on snow depth from Boike et al. (2018).

..., the snow-free period 138 ± 18 days. Snow depth was reported by Boike et al. (2018) averaging 0.3 m between 2002 and 2017 with a maximum of 0.8 m in 2017. Beginning in early to mid-June,...

RC 12

p.5, l.6ff: you may add the power consumption as another important difference between CP and OP systems.

I added a remark on power consumption to the sentence starting in line 11 of page 5.

OP sensors are commonly installed in close proximity to the anemometer and do not require a pump that greatly reduces the power consumption of OP instruments compared to CP setups.

RC 13

p.6, l.4ff: even though you spend a few sentences to describe the WPL-approach, you fail to mention that this is about accounting for the influence of density fluctuations

I agree, the WPL-approach needs a more thorough and clear introduction. I therefore rewrote the section from page 5, line 14 (starting with "CP analyzers have the...") until page 6, line 8 (before "Major drawbacks...") and moved it to a new paragraph.

Infrared gas analyzers typically measure gas densities and report the number of molecules per volume of air. To be able to refer the mass of a gas to the mass of air, gas densities are transformed to mixing ratios using air density. However, as the optical path of an OP gas analyzer is exposed to the varying temperature, pressure and humidity conditions of the atmosphere, air density in the measurement cell fluctuates mainly due to thermal expansion/contraction and water dilution/concentration. This effect, that leads to faulty concentration readings of OP instruments and thereby to incorrect flux estimates, has first been described by Webb et al. (1980). The authors proposed two flux correction terms to compensate for these density fluctuation effects that are referred to as Webb-Pearman-Leuning (WPL) terms and have since been verified experimentally and theoretically and are routinely applied in OP EC studies. Especially at times of low gas fluxes, WPL terms can become orders of magnitude larger than raw gas fluxes (Munger et al., 2012). CP analyzers have the advantage of controlled temperature and pressure conditions in the measurement cell, allowing for the sample-wise calculation of mixing ratios rather than molar densities (Ibrom et al., 2007b) and thereby avoiding the need to apply air density fluctuation correction terms after raw flux calculation.

RC 14

p.7, Section 3.3: It's a bit odd that you start describing some elements of quality flagging already in Section 3.2, and continue with this material here, in the main quality section. This should be cleaned up. Also, you fail to reference Table 3 in the text. Moreover, you should improve the structure of this Section. You begin with a too short general overview on additional quality filters, and how they are used in the overall QC flagging scheme. You then close the section with very similar statements. This should be merged to a single introductory paragraph that clearly states that you applied 6 more quality checks, and if any of them indicated problems, the quality flag was set to 2.

Thank you for the suggestion, I agree, the "Quality filtering" section (3.3) should be more clear. I moved the end of section 3.2 (from page 7 line 21 to the end of the paragraph) to a newly formulated introduction of section 3.3. As suggested, I also moved the end of section 3.3 (page 8, line 23, starting from "In the dataset available...") into this new introductory paragraph. Section 3.3 now begins with:

We set EddyPro to calculate quality flags according to Mauder and Foken (2004) that represent flux quality in three classes (0, 1 and 2) with 0 denoting the highest and 2 denoting the lowest quality class. This quality evaluation is based on tests for stationarity and developed turbulence and thereby indicates whether general EC assumptions about atmospheric conditions were met during a flux calculation period. Flux quality assessment was largely based on the scheme of Mauder and Foken (2004). In the data set available for download, we included one column for each analyzer type containing this quality flag. Additionally, we applied six further screening steps and flagged fluxes of low quality. If a flagged flux was not already assigned to class 2 according to Mauder and Foken (2004), we set the quality flag to 2. Fluxes of quality class 2 should be omitted from further analysis. They are included in the reported dataset for the sake of completeness. We performed the six additional flagging steps in the following

sequence. An overview of these filtering steps including the number of flagged values is given in Table 3 (*in original draft*).

RC 15
p.8, l.14: The choice of 450ppm as the upper concentration limit seems rather narrow. Can you please justify?
I want to stress that this limit refers to half-hourly average concentrations, the absoute concentration filter applied to the high frequency data during raw data screening in EddyPro (following Vickers & Mahrt, 1997) allowed a much wider range (200 ppm to 900 ppm) of concentrations. The limit of half-hourly average concentrations was decided for after calculating the 95[th] percentile of closed-path (440 ppm) and open-path (410 ppm) averages for timesteps with flux qualities 0 and 1.

RC 16
p.9, Fig.2: Figure 2 isn't really informative, since it's hard to distinguish between corrected and uncorrected time series in such a cloud of values. Please think about a different format (box plots?), or just leave out the plots, and show the regression statistics instead in a table.
I replaced the figure and added a table with the regression statistics.

[Figure]

Fig. 7    Effect of the self-heating correction on the correlation between open-path (OP) and closed-path (CP) fluxes (left panel). Only quality class 0 is shown. Negative fluxes are affected more strongly by the correction than positive fluxes (right panel).

Table 1   Spearman's rank correlation coefficient rs and Pearson's correlation coefficient r between closed-path (CP) and open-path (OP) fluxes with and without the applied self-heating correction. The agreement between CP and OP fluxes increases throughout all quality classes after OP correction.

|  |  | Quality class 0 | Quality classes 0,1 | Quality classes 0, 1, 2 |
|---|---|---|---|---|
| $r_s$ | OP uncorrected | 0.896 | 0.866 | 0.508 |
|  | OP corrected | 0.907 | 0.871 | 0.512 |
| $r$ | OP uncorrected | 0.894 | 0.871 | 0.042 |
|  | OP corrected | 0.904 | 0.877 | 0.055 |

RC 17
p.10, Section 3.5: I suppose Figs. 3 & 4 should belong to this section. They are not  referred to in the text. Moreover, it's not necessary to show Fig.4, since given the minor absolute shifts in fluxes after Burba correction in this case, the differences between  figures are not discernible. As an alternative for Fig.4, it may be interesting to show the gap-filled time series, maybe even in cumulative form?

I agree, the gain of information from Figures 3 and 4 in the original draft is limited. I replaced both with one new Figure (Fig. 2 in this document) showing the measured time series that we compiled from open-path and closed-path records as well as the gap-filled time series.

[Figure]

Fig. 8    Multiannual carbon dioxide flux time series compiled from fluxes measured with closed-path and open-path sensors on Samoylov Island's river terrace. Fluxes of quality class 2 are not shown. Self-heating errors in the OP data set have been corrected for. Additionally, the result from gap filling this time series with the MDS method is shown. The given numbers of values for the gap-filled time series include measured fluxes.

RC 18
p.11, Section 3.6: while the method applied to calculate footprints is sufficiently detailed,  it is not fully clear how footprint results were combined with the land cover map.  What's completely missing here is a reference to the findings, a.k.a. a bottom line. As  already mentioned in the 'medium comments' above, this is an important piece of information,  since (as shown in Table 1) multiple positions with multiple sensor heights were  used over the 16 year data record. The authors clearly need to point out that this mixture  of setups is still suitable to form a coherent, long-term time series of flux exchange  for this site. It's not sufficient to just briefly mention these results in the conclusions.  In particular, the results in Table 5 emphasize that the southernmost tower position,  used within the years 2007-2009, featured a quite different composition of landscape  elements than the northern site position. The authors need to make an effort to convince  the readers that these differences did not result in a significant deviation of flux  patterns, and therefore would bias the long-term trends.

I added a new "Discussion" section detailing the effects of tower relocations. See my response to RC 2 above.

I added more information on how the footprint results were combined with the land cover map to the end of section "Footprint modeling"

...We evaluated the footprint model at the same resolution that was used by Muster et al. (2012) to classify the surface (i. e. 0.14 m x 0.14 m). We could thereafter assign a probability of being the EC source area to each classified pixel and sum up the probabilities of all pixels belonging to the same surface class to estimate the contribution of each class. This proceeding to combine an EC source area estimation with a land cover classification is similar to what has been applied and described in more detail by Forbrich et al. (2011).

RC 19

p.11, Section 4: It's good to list the parameters given in the PANGAEA dataset in a separate table. However, since this dataset is obviously restricted to CO2 fluxes and their QC parameters, it would be good to also list the source for ancillary meteorological information, if available, since those will be necessary to put the flux time series into contex

I added a reference to ancillary measurements to the "Data availability section.

Ancillary long-term time series of meteorological and soil variables from Samoylov Island are available from Boike et al. (2018) and can be accessed through https://doi.pangaea.de/10.1594/PANGAEA.891142

I also added a new paragraph to the conclusions pointing out the importance of these ancillary data.

Furthermore, analysis of this NEE time series is not limited to the gas flux data only. An extensive data stream of meteorological and soil variables between 2002 and 2017 has recently been published by Boike et al. (2018). The authors made their records publicly accessible on the two long-term repositories Pangaea (https://doi.pangaea.de/10.1594/PANGAEA.891142) and Zenodo (https://zenodo.org/record/2223709). The fact of parallelly available ancillary ecosystem variables enables a potential user to put the gas flux dynamics reported in this publication into context with the variability of other ecosystem properties and potential flux drivers. We regard this type of analysis as vital to understand inter-annual variability of CO2 fluxes on Samoylov Island and are working on it ourselves (Kutzbach, unpublished).

New References

Chadburn, S. E., Krinner, G., Porada, P., Bartsch, A., Beer, C., Belelli Marchesini, L., Boike, J., Ekici, A., Elberling, B., Friborg, T., Hugelius, G., Johansson, M., Kuhry, P., Kutzbach, L., Langer, M., Lund, M., Parmentier, F.-J. W., Peng, S., Van Huissteden, K., Wang, T., Westermann, S., Zhu, D., and Burke, E. J.: Carbon stocks and fluxes in the high latitudes: using site-level data to evaluate Earth system models, Biogeosciences, 14, 5143–5169, 2017.

Eckhardt, T., Knoblauch, C., Kutzbach, L., Simpson, G., Abakumov, E., and Pfeiffer, E.-M.: Partitioning CO2 net ecosystem exchange fluxes on the microsite scale in the Lena River Delta, Siberia, Biogeosciences Discussions, 2018, 1–27, 2018.

Porada, P., Weber, B., Elbert, W., Pöschl, U., and Kleidon, A.: Estimating global carbon uptake by lichens and bryophytes with a processbased model, Biogeosciences, 10, 6989–7033, 2013.

Raddatz, T., Reick, C., Knorr, W., Kattge, J., Roeckner, E., Schnur, R., Schnitzler, K.-G., Wetzel, P., and Jungclaus, J.: Will the tropical land biosphere dominate the climate–carbon cycle feedback during the twenty-first century?, Climate Dynamics, 29, 565–574, 2007.

Richardson, A. D., Aubinet, M., Barr, A. G., Hollinger, D. Y., Ibrom, A., Lasslop, G., and Reichstein, M.: Uncertainty quantification, in: Eddy Covariance: A Practical Guide to Measurement and Data Analysis, edited by Aubinet, M., Vesala, T., and Papale, D., pp. 173–209, Springer, 2012.

Richardson, A. D., Hollinger, D. Y., Burba, G. G., Davis, K. J., Flanagan, L. B., Katul, G. G., Munger, J. W., Ricciuto, D. M., Stoy, P. C., Suyker, A. E., et al.: A multi-site analysis of random error in tower-based measurements of carbon and energy fluxes, Agricultural and Forest Meteorology, 136, 1–18, 2006.

Rinne, J., Douffet, T., Prigent, Y., and Durand, P.: Field comparison of disjunct and conventional eddy covariance techniques for trace gas flux measurements, Environmental pollution, 152, 630–635, 2008.

Dragoni, D., Schmid, H. P., Grimmond, C. S. B., and Loescher, H. W.: Uncertainty of annual net ecosystem productivity estimated using eddy covariance flux measurements, Journal of Geophysical Research: Atmospheres, 112, 2007.

---

## Author Comment (AC2) · 1 Feb 2019

Author reply to Referee comments from **Anonymous Refferee # 2** from 29 November 2018 (https://doi.org/10.5194/essd-2018-98-RC2, 2018 ) on:

**A long-term (2002 to 2017) record of closed-path and open-path eddy covariance CO$_2$ net ecosystem exchange fluxes from the Siberian Arctic**
by David Holl et al.

Reviewer comments (RC)
Author comments (AC)
Manuscript changes (MC)
Please note that figure numbers in this document refer to figures in this document and not to the numbering in the original draft unless denoted otherwise.

RC 1
Review ESSD-2018-98 Siberian Permafrost
A login barrier on Pangaea prevents me from downloading the actual .tsv files. As it turns out I have a valid Pangaea login but, once in, I still cannot access the data. THIS VIOLATES ESSD POLICIES AND PREVENT ME OR ANY OTHER USER FROM FULL EVALUATION OF THE DATA!!
This must be fixed immediately. All my comments below assume a quality effort on the part of the authors but until the data becomes fully and freely accessible, I must withhold approval of this manuscript.
The dataset is now accessible without limitations (https://doi.pangaea.de/10.1594/PANGAEA.892751). I addressed this issue with a short comment during the interactive discussion (SC1: 'Data access', David Holl, 14 Dec 2018).

RC 2
Page 3 line 15: this sentence should refer to eddy covariance "systems" because more than one instrument type was deployed on three different towers at two different locations.
I agree, "system" was changed to "systems".

RC 3
Page 3 line 30: technically, "low evaporation rates" should lead to dryer than expected atmospheric humidities. Perhaps the authors refer here to soil (moisture) conditions or to a combination of low specific atmospheric water vapour contents with lower temperatures that lead to a relatively high relative humidity?
The point we wanted to make here is that even though water input by precipitation is not very high on an annual basis, water loss by evaporation is even smaller. We failed to mention that low evaporation is indeed connected to low ambient temperatures and low water vapour pressure deficits.
I changed the sentence to:
An arctic-continental climate with low mean annual temperatures prevails in the Lena River Delta. Although precipitation is low as well, the climate can be considered humid as evaporation rates are low due to low ambient temperatures and relative humidity is high.

RC 4
Page 4, soils: much higher resolution soil mapping documented here than in the northern circumpolar soil atlas (Jones et al. 2010) so I understand better details here. But Jones et al. soil atlas, at least for central Siberia, adopts the "Russian Soil Classification System" while this paragraph references US or FAO definitions. Why? Because permafrost carbon estimates (e.g. Hugelius et al. in ESSD 2013, not cited here but used extensively in Koven and Schuur, both of which these authors do cite) depend substantially and in fact influence soil type classifications (e.g. again see Hugelius) in permafrost regions, this paper should at least document consistency with other soil classification systems? I know this is not a description of Samoylov permafrost soils data set, but we should at least know consistency or valid reasons for inconsistencies between soil classifications systems used by flux vs soil carbon communities?

The "Site description" section collects data that has been reported from Samoylov in the past. To my knowledge, Jones et al. 2010 do in fact use the FAO WRB system for soil classification. Unfortunately, Pfeiffer and Grigoriev (2002) and Zubrzycki et al. (2013) used a different classification. Hugelius et al. 2014 ("Estimated stocks of circumpolar permafrost carbon with quantified uncertainty ranges and identified data gaps", Biogeosciences) did use SOC data provided by Zubrzycki et al. (2013) and therefore appear to regard this dataset as consistent with the methods and resulting estimates from other authors.

RC 5
Page 4 line 20 "contributied"
I corrected this typo.

RC 6
Page 4 line 25 (and following): "wind speed" I think you actually mean wind velocity because, unlike speed, you need both magnitude and direction?
I agree. I replaced "wind speed" with "wind velocity" at the appropriate places.

RC 7
Page 4 line 28: Need to mention here that, for a period of two years, the tower location moved almost 1 km to the west-southwest?
I think this fact is already made clear by saying "three different tower structures" and referring to Figure 1 (*in the original draft*) where the different locations are illustrated. The impacts of tower relocation are disscussed in more depth than in the original manuscript in a newly added "Disscussion" section. See my reply to RC 14.

RC 8
Page 4 line 30: For one year sampling rate went to 10 Hz, and later for a period of roughly 9 months sampling went to 5 Hz. Opportunity to test sampling and influence on spectral properties of flux calculations?
I agree, we missed the opportunity to report the impact of changing sampling frequencies on the spectral properties of the raw data. I added a new Discussion section to the manuscript where this topic is addressed. See my reply to RC 14.

RC 9
Page 5 line 2: "... the data set contains year-round fluxes in some years ...". But, from Figure 3, only 2016 had anything close to full annual coverage (e.g. roughly 10k valid 0.5-hour observations out of a maximum possible of 17.5k). No other year shows anything close to full four-season data coverage. If the authors contend that 2014 and 2010 also provide "year-round" flux data then they have a very low standard/expectation for what constitutes valid year-round performance which they should share with readers.
I agree, the cited statement is a bit vague. I replaced
"Although data coverage is biased towards the growing season, the data set contains year-round fluxes in some years (see Table 1)."
with
Although data coverage is biased towards the growing season, the dataset contains considerably more shoulder season and winter fluxes in its second half from 2010 to 2017(see Table 1 *(in original draft)*). The also increasing availability of year-round ancillary meteorological data resulted in gap filled flux time series covering each half hour in the two years 2010 and 2016 (see Figure 1).
To illustrate data coverage better than only in Table 1 , I added a new figure containing corrected measured and gap-filled fluxes.

[Figure]

gap-filled **measured**

$F_{CO_2}$, µmol m$^{-2}$ s$^{-1}$

| 2002 | n = 2564 | **n = 1794** |
| 2003 | n = 4557 | **n = 3211** |
| 2004 | n = 2569 | **n = 1731** |
| 2005 | n = 2196 | **n = 1834** |
| 2006 | n = 5087 | **n = 3917** |
| 2007 | n = 2011 | **n = 1431** |
| 2008 | n = 7571 | **n = 3634** |
| 2009 | n = 9455 | **n = 3995** |

| 2010 | n = 17520 | **n = 7980** |
| 2011 | n = 11197 | **n = 5613** |
| 2012 | n = 5748 | **n = 3208** |
| 2013 | n = 8867 | **n = 4290** |
| 2014 | n = 12011 | **n = 7329** |
| 2015 | n = 11495 | **n = 8063** |
| 2016 | n = 17568 | **n = 11321** |
| 2017 | n = 13096 | **n = 8973** |

Fig. 1     Multiannual carbon dioxide flux time series compiled from fluxes measured with closed-path and open-path sensors on Samoylov Island's river terrace. Fluxes of quality class 2 are not shown. Self-heating errors in the OP dataset have been corrected for. Additionally, the result from gap filling this time series with the MDS method is shown. The given numbers of values for the gap-filled time series include measured fluxes.

RC 10
Page 6 line 4: "These Webb-Pearman-Leuning (WPL, Webb et al., 1980) terms". 'These' in this case refers to terms necessary to calculate density for OP measurements but sentence as written allows confusion. Better to specify temperature, pressure, water vapour content, etc.
RC 11
Page 6 line 7: "undisturbed heat fluxes". Reviewer may know what you mean by 'undisturbed' and why you need those, but you have not explained clearly to readers.
RC 12
Page 6 line 7: "WPL terms" By this point authors should have told readers exactly what they mean when they say 'WPL terms'. Jargon creeps in here, as well as assumption that every reader already knows the intricacies of eddy correlation measurements. Not true! Please rewrite the initial sentences of this paragraph in a clearer form and format.

Referring to comments RC 10 to RC 12: I agree, the section on the corrections for air density fluctuations are somewhat confusing and too brief. I therefore rewrote the section from page 5, line 14 (starting with "CP analyzers have the…") until page 6, line 8 (before "Major drawbacks…") and moved it to a new paragraph in an effort to include a more understandable and thorough description of the WPL approach.

Infrared gas analyzers typically measure gas densities and report the number of molecules per volume of air. To be able to refer the mass of a gas to the mass of air, gas densities are transformed to mixing ratios using air density. However, as the optical path of an OP gas analyzer is exposed to the varying temperature, pressure and humidity conditions of the atmosphere, air density in the measurement cell fluctuates mainly due to thermal expansion/contraction and water dilution/concentration. This effect, that leads to faulty concentration readings of OP instruments and thereby to incorrect flux estimates, has first been described by Webb et al. (1980). The authors proposed two flux correction terms to compensate for these density fluctuation effects that are referred to as Webb-Pearman-Leuning (WPL) terms and have since been verified experimentally and theoretically and are routinely applied in OP EC studies. Especially at times of low gas fluxes, WPL terms can become orders of magnitude larger than raw gas fluxes (Munger et al., 2012). CP analyzers have the advantage

of controlled temperature and pressure conditions in the measurement cell, allowing for the sample-wise calculation of mixing ratios rather than molar densities (Ibrom et al., 2007b) and thereby avoiding the need to apply air density fluctuation correction terms after raw flux calculation.

RC 13
Why does a reader find Figs 1 and 2 introduced at appropriate points within the text but all Tables and Figs 3 and 4 appearing at the end of the manuscript after text and references. Need to fix this now and check it again during proofreading.

Figures 3 and 4  (numbering of discussion paper) were removed. See my reply to RC 15 below.
The table placement after the references is part of the ESSD guidelines for manuscript preparation.
On the website (https://www.earth-system-science-data.net/for_authors/manuscript_preparation.html) it says: "Any tables should appear on separate sheets after the references and should be numbered sequentially with Arabic numerals."
I suspect that typesetting of the final version by the publisher will result in a placement of tables closer to the references to it in the text.

RC 14
The authors' descriptions of data processing, quality filtering, self-heating corrections, temporal gap filling etc. seem appropriate and well-described. However, we find no assessment of the the two-year period of tower relocation. Mentioned in the introduction and again in the conclusions, but completely absent from the data processing and data quality descriptions. If that relocation does not matter, e.g. had no effect on time series or data quality, then readers must question the footprint analysis, as mentioned in the Conclusion! Based on lack of information here, this statement from the conclusion " ... ensuring that EC source area deviations are quantifiable by a potential user" seems unsupportable for at least two years? One also wonders about the earlier documentation that sampling frequencies changed (e.g. 20 Hz to 10 Hz to briefly 5 Hz). Did those changes also have no effect (or no utility) on data processing. The authors seem to expect users to ignore these possibly substantial location sampling issues but, having mentioned both changes (good) they then fail to report corrections or consequences (bad).

I added a new "Discussion" section to the manuscript addressing these issues.
Although we did our best to ensure the consistency and appropriateness of the data processing workflow for the presented NEE time series, due to technical and logistical constraints during 16 years of field work, disparities in the experimental setup exist which may challenge its integrity. The EC tower was relocated twice, the measurement height was changed three times (see Figure 1  and Table 1 *(in original draft)*). These changes of tower location and measurement height affected the source area and hence the surface types sampled during flux measurements. Most notably, between July 2007 and June 2009, the EC tower was placed about 650 m south-west of its original position at the center of Samoylov Island, in an area with an increased coverage of the surface class *wet tundra*. This is revealed by the footprint analysis (Figure 2). While the EC footprint is dominated by the surface class *dry tundra* throughout the time series, during subperiods 2007, 2008 and 2009 I the contributions of *wet tundra* to the measured flux are significantly higher.
To check the effect of the shifts in tower location and measurement height on cumulative $CO_2$-C fluxes, we calculated flux sums for a period when flux time series without gaps were available in most years. The overlapping period covers days of year 200 to 234, i.e. part of the growing season in all years except for 2004 (see Figure 3). Interannual variability of cumulative C fluxes in years with constant tower location (and measurement height) appears to be large and driven by a more complex set of variables than shifts in surface class contributions only. Flux sums from the periods when EC tower relocation led to a significant shift in EC footprint composition are well within the range of the distribution of cumulated fluxes from years with a more homogeneous EC fetch area. We therefore assume that, at least with respect to budget calculations, the presented long-term time series is not disrupted and can be regarded as representative for a polygonal tundra site dominated by dry tundra. For a more in depth analysis of flux dynamics, footprint information should and can be considered by users of the data set. Recently, a comparison between surface class level NEE models based on chamber measurements with EC fluxes, using the half-hourly footprint information provided in this data set for scaling, yielded good agreement between the results obtained with both methods Eckhardt et al. (2018). We regard the availability of half-hourly footprint information in the presented NEE data set an attribute that sets it apart from other studies and holds chances for comprehensive analyses.
Apart from the changes in anemometer height, other deviations of the general instrument setup occurred due to limitations in data storage during two winter periods when the acquisition frequency was reduced to 5 Hz

and 10 Hz respectively. Rinne et al. (2008) demonstrated in a field experiment that fluxes calculated from raw data recorded at frequencies below 20 Hz compare well with fluxes derived from high frequency raw data. Differences arise as an increase of random noise and not as a systematic bias. High frequency noise removal before ensemble spectra estimation in EddyPro is effective in limiting the effect of increased noise on the quality of transfer function estimation in the process of spectral correction. Overall spectral correction in EddyPro is expressed as a spectral correction factor (SCF) which comprises the effect of all applied compensations for high and low frequency loss. Raw fluxes are multiplied with the respective SCFs during processing. We compared the SCF distributions of the two above mentioned winter periods with statistics of the remaining parts of the time series when data was recorded at 20 Hz. SCF deviations between the different acquisition frequencies are minor (see Figure 4) implying that systematic differences between fluxes calculated form raw data of different temporal resolutions are in fact small, random uncertainties increase, however.

[Figure]

Fig. 2    Mean surface class composition of the eddy covariance footprint during 17 subperiods of four different tower setups at three locations on Samoylov Island.

[Figure]

Fig. 3     Comparison of cumulative CO2 flux sums of different years during the same day of year range.

[Figure]

Fig. 4     Spectral correction factor statistics for periods with different acquisition frequencies.

RC 15
Fig. 4 not referenced in text? In the (barely viewable) version provided for review, Figure 3 and Figure 4 look identical, even to having identical numbers (n) of samples. The legends for the two figures differ slightly, but the figures themselves differ not at all. Wrong figure in the wrong place? Or, because we find no mention of Figure 4 in the text, one figure wrongly duplicated? Serious error, needs attention.

There is indeed a difference between Figures 3 and 4 (note that these figure numbers refer to the original draft, not the figures we see above, on this page). It is, however, fairly subtle. The difference between the plots only refers to OP fluxes, Figure 3 shows fluxes that are corrected for the self-heating effect, Figure 4 shows uncorrected fluxes, hence the number of samples are identical. My intent was for a reader to be able to flip back and forth between pages 19 and 20 in the pdf file viewed in full-screen on a computer to get a visual impression of the impact of the self-heating correction on OP fluxes. This intention clearly did not pan out, I will therefore remove Figures 3 and 4 and replace them with a new Figure (Figure 1 in this document) showing corrected measurement data and gap filled fluxes for a better overview of the final time series.

RC 16

Analysis and use of this particular CO2 data will require simultaneous access to observation time series from boreholes, river gauges, water sampling, long-term meteorology, e.g. Boike et al. 2018. Very important to connect these two data sets. If Boike et al. emerges successfully from ESSD, then we need a close explicit link described here. Until Boike et al. appear in ESSD or elsewhere, release of this data seems premature at best. If other sources of necessary soil, radiation, micrometeorlogical, etc. data exist, please reference those as well or instead? Tiksi?

I agree, the fact that long-term meteorological and soil records are available makes our NEE time series much more valuable. Boike et al. (2018) archived their data in two long-term repositories (Pangaea and Zenodo). Links are added in the following paragraph that I added to the end of the conclusion.

Furthermore, analysis of this NEE time series is not limited to the gas flux data only. An extensive data stream of meteorological and soil variables between 2002 and 2017 has recently been published by Boike et al. (2018). The authors made their records publicly accessible on the two long-term repositories Pangaea (https://doi.pangaea.de/10.1594/PANGAEA.891142) and Zenodo (https://zenodo.org/record/2223709). The fact of simultaneously available ancillary ecosystem variables enables a potential user to put the gas flux dynamics reported in this publication into context with the variability of other ecosystem properties and potential flux drivers. We regard this type of analysis as vital to understand inter-annual variability of $CO_2$ fluxes on Samoylov Island and are working on it ourselves (Kutzbach, unpublished).

RC 17

Note reference to methane measurements. Do these authors consider that they have now have sufficient information, biogeochemical and ecological, to construct an annual carbon budget? If so, they should at least assure readers of forthcoming analyses. If not, why not? Lack of winterseason measurements? Can only construct a valid annual budget for, e.g., 2016?

We regard this dataset publication as a starting point for analysis of flux dynamics done by us and other members of the scientific community. Yes, we think the literature record of information necessary to conduct carbon budget calculations for Samoylov Island has certainly grown over the last years. Especially the manuscript at hand contributes an important part of the information necessary to tackle the task of budget calculations successfully. As mentioned before, we are aiming at publishing those types of results in the future (Kutzbach, unpublished). In this paper, however, we wanted to focus on the methods we used to process the data rather than its interpretation. To our understanding, this proceeding is in line with the "Aims and scope" of ESSD, which is one reason why we selected this journal.

*"Articles in the data section may pertain to the planning, instrumentation, and execution of experiments or collection of data. Any interpretation of data is outside the scope of regular articles. Articles on methods describe nontrivial statistical and other methods employed (e.g. to filter, normalize, or convert raw data to primary published data) as well as nontrivial instrumentation or operational methods. Any comparison to other methods is beyond the scope of regular articles."* *(https://www.earth-system-science-data.net/about/aims_and_scope.html)*

RC 18

What makes this time series interesting? Why not simply download from FLUXNET2015? Presumably this data serves as important piece of the tundra carbon analysis presented by Zono et al.? For this reader, the authors have not made an adequate case about potential importance and utility. Readers might consider it potentially very important but this statement, again from the conclusion - "a valuable addition to the already existing data base of CO2 net ecosystem exchange observations from the Arctic" - seems weak and vague. Do the authors claim to have produced a unique high-quality data set or just another contribution to FLUXNET. If the former, then ESSD seems an appropriate venue. If the latter, why bother? Publish the entire FLUXNET data set instead? Again, this reader favours the former but the authors have not made a strong case.

We are currently talking with FLUXNET staff about how to submit our updatad time series. Some of the information we provide is beyond what one could find in a typical FLUXNET release. At this time, I am not sure if our parallel data stream of open and closed-path sensors from a single site conforms to the FLUXNET data format. For the same reason, the EC footprint data will most likely stay exclusive to the Pangaea release of this data set. While shifts in EC footprint composition are important to understand flux dynamics they are not routinely included in FLUXNET data sets. Besides the latter fact that emphasizes the uniqueness of the presented data set, I think a publication in ESSD is further justified by the revised analysis of the overall time

series consistency with respect to instrumentation and location changes as well as by the thorough description of data processing/quality filtering. Furthermore, a detailed description of site characteristics including the relevant literature put the data set into context beyond of what is possible within a FLUXNET release.

New References

Chadburn, S. E., Krinner, G., Porada, P., Bartsch, A., Beer, C., Belelli Marchesini, L., Boike, J., Ekici, A., Elberling, B., Friborg, T., Hugelius, G., Johansson, M., Kuhry, P., Kutzbach, L., Langer, M., Lund, M., Parmentier, F.-J. W., Peng, S., Van Huissteden, K., Wang, T., Westermann, S., Zhu, D., and Burke, E. J.: Carbon stocks and fluxes in the high latitudes: using site-level data to evaluate Earth system models, Biogeosciences, 14, 5143–5169, 2017.

Eckhardt, T., Knoblauch, C., Kutzbach, L., Simpson, G., Abakumov, E., and Pfeiffer, E.-M.: Partitioning CO2 net ecosystem exchange fluxes on the microsite scale in the Lena River Delta, Siberia, Biogeosciences Discussions, 2018, 1–27, 2018.

Porada, P., Weber, B., Elbert, W., Pöschl, U., and Kleidon, A.: Estimating global carbon uptake by lichens and bryophytes with a processbased model, Biogeosciences, 10, 6989–7033, 2013.

Raddatz, T., Reick, C., Knorr, W., Kattge, J., Roeckner, E., Schnur, R., Schnitzler, K.-G., Wetzel, P., and Jungclaus, J.: Will the tropical land biosphere dominate the climate–carbon cycle feedback during the twenty-first century?, Climate Dynamics, 29, 565–574, 2007.

Richardson, A. D., Aubinet, M., Barr, A. G., Hollinger, D. Y., Ibrom, A., Lasslop, G., and Reichstein, M.: Uncertainty quantification, in: Eddy Covariance: A Practical Guide to Measurement and Data Analysis, edited by Aubinet, M., Vesala, T., and Papale, D., pp. 173–209, Springer, 2012.

Richardson, A. D., Hollinger, D. Y., Burba, G. G., Davis, K. J., Flanagan, L. B., Katul, G. G., Munger, J. W., Ricciuto, D. M., Stoy, P. C., Suyker, A. E., et al.: A multi-site analysis of random error in tower-based measurements of carbon and energy fluxes, Agricultural and Forest Meteorology, 136, 1–18, 2006.

Rinne, J., Douffet, T., Prigent, Y., and Durand, P.: Field comparison of disjunct and conventional eddy covariance techniques for trace gas flux measurements, Environmental pollution, 152, 630–635, 2008.

Dragoni, D., Schmid, H. P., Grimmond, C. S. B., and Loescher, H. W.: Uncertainty of annual net ecosystem productivity estimated using eddy covariance flux measurements, Journal of Geophysical Research: Atmospheres, 112, 2007.